# Flexibility of intrinsically disordered degrons in AUX/IAA proteins reinforces auxin co-receptor assemblies

Michael Niemeyer [1], Elena Moreno Castillo[1], Christian H. Ihling[2], Claudio Iacobucci[2], Verona Wilde[1], Antje Hellmuth[1], Wolfgang Hoehenwarter[3], Sophia L. Samodelov [4], Matias D. Zurbriggen [4], Panagiotis L. Kastritis[5], Andrea Sinz [2] & Luz Irina A. Calderón Villalobos [1✉]

Cullin RING-type E3 ubiquitin ligases SCF[TIR1/AFB1-5] and their AUX/IAA targets perceive the phytohormone auxin. The F-box protein TIR1 binds a surface-exposed degron in AUX/IAAs promoting their ubiquitylation and rapid auxin-regulated proteasomal degradation. Here, by adopting biochemical, structural proteomics and in vivo approaches we unveil how flexibility in AUX/IAAs and regions in TIR1 affect their conformational ensemble allowing surface accessibility of degrons. We resolve TIR1·auxin·IAA7 and TIR1·auxin·IAA12 complex topology, and show that flexible intrinsically disordered regions (IDRs) in the degron's vicinity, cooperatively position AUX/IAAs on TIR1. We identify essential residues at the TIR1 N- and C-termini, which provide non-native interaction interfaces with IDRs and the folded PB1 domain of AUX/IAAs. We thereby establish a role for IDRs in modulating auxin receptor assemblies. By securing AUX/IAAs on two opposite surfaces of TIR1, IDR diversity supports locally tailored positioning for targeted ubiquitylation, and might provide conformational flexibility for a multiplicity of functional states.

[1] Molecular Signal Processing Department, Leibniz Institute of Plant Biochemistry (IPB), Weinberg 3, 06120 Halle (Saale), Germany. [2] Department of Pharmaceutical Chemistry & Bioanalytics, Institute of Pharmacy, Martin Luther University Halle-Wittenberg, Charles Tanford Protein Center, Kurt-Mothes-Straße 3a, 06120 Halle (Saale), Germany. [3] Proteome Analytics, Leibniz Institute of Plant Biochemistry (IPB), Weinberg 3, 06120 Halle (Saale), Germany. [4] Institute of Synthetic Biology & Cluster of Excellence on Plant Science (CEPLAS), Heinrich-Heine University of Düsseldorf, Universitätsstrasse 1, 40225 Düsseldorf, Germany. [5] ZIK HALOMEM & Institute of Biochemistry and Biotechnology, Martin Luther University Halle-Wittenberg, Biozentrum, Weinbergweg 22, 06120 Halle (Saale), Germany. ✉email: LuzIrina.Calderon@ipb-halle.de

Proteolysis entails tight spatiotemporal regulation of cellular protein pools[1,2]. The ubiquitin-proteasome system (UPS) rules over protein turnover, and controls stimulation or attenuation of gene regulatory networks through transcriptional repressors or activators[2]. Typical E1-E2-E3 enzymatic cascades warrant specific ubiquitylation by catalyzing the ATP-dependent attachment of ubiquitin moieties to target proteins[3]. Directly and indirectly, every single aspect of cellular integrity and adaptation is impacted by protein ubiquitylation, e.g., cell cycle progression, apoptosis/survival, oxidative stress, differentiation, and senescence[4].

In SKP1/CULLIN1/F-BOX PROTEIN (SCF)-type E3 ubiquitin ligases, the interchangeable F-box protein (FBP) determines specificity to the E3 through direct physical interactions with the degradation targets[5,6]. These carry a short degradation signal or degron, located mostly within structurally disordered regions, which is recognized by cognate E3 ligases[7]. Primary degrons within a protein family, whose members share the same fate, behave as islands of sequence conservation surrounded by fast divergent intrinsically disordered regions (IDRs)[7]. Once a favorable E3-target association stage is accomplished, one or multiple lysine (Lys) residues in the target become accessible[8–10]. Conformational flexibility on the part of the E3-target ensemble permits then an E2-loaded with Ub (E2~Ub) to approach the bound target, such that a suitable microenvironment for catalytic Ub transfer is created[7]. Efficient degradation by the UPS requires the 26S proteasome to bind its protein target through a polyubiquitin chain with a specific topology, and subsequently engages the protein at a flexible initiation region for unfolding and degradation[11]. A primary degron for E3 recruitment, a ubiquitin chain on specific Lys residues, together with IDRs are the basic elements for efficient ubiquitin-mediated proteasomal degradation[7].

Biological active intrinsically disordered proteins (IDPs) and IDRs exist as structural non-uniform ensembles, due to dynamic back-bone movement[12]. Some functions of IDPs are entropic in nature and originate precisely from their lack of well-defined structure[13]. UPS targets often contain IDRs or are IDPs functioning, i.e., in plant signal transduction[14–17]. The auxin indole 3-acetic acid (IAA) promotes plant growth and development by triggering the degradation of auxin/indole-3-acetic-acid proteins (AUX/IAAs), which leads to changes in gene expression[18]. AUX/IAAs are mostly short-lived transcriptional repressors with half-lives varying from ~6 to 80 min, and the expression of most family members is rapidly (<15 min) induced by auxin[19]. The *Arabidopsis* genome encodes for 29 AUX/IAAs, and 23 of them carry a mostly conserved VGWPP-[VI]-[RG]-x(2)-R degron as recognition signal for an SCF$^{TIR1/AFB1-5}$ E3 ubiquitin ligase for auxin-mediated AUX/IAA ubiquitylation and degradation[20,21]. Under low auxin concentrations, AUX/IAAs are stabilized and repress type A ARF (auxin response factor) transcription factors via physical heterotypic interactions through their type I/II Phox/Bem1p (PB1) domain (formerly known as DIII-DIV) and recruitment of topless (TPL) co-repressors[21,22]. When auxin levels reach a certain threshold, FBPs transport inhibitor response 1 (TIR1)/auxin signaling F-box 1–5 (AFB1-5) gain affinity for the AUX/IAA degron by direct IAA binding[23,24]. The resulting AUX/IAA ubiquitylation and degradation ensues ARF derepression and auxin-induced transcriptional changes[25]. Since AUX/IAA transcripts are themselves auxin regulated, they act, once the intracellular AUX/IAA pool is replenished, in a negative feedback loop repressing ARF activity de novo[26,27].

These molecular interactions establish highly pleiotropic and complex physiological and morphological auxin responses during plant development[28]. During embryogenesis for instance, auxin controls normal organ formation, as evidenced by early developmental arrest in several auxin response mutants[29]. Loss of ARF5 function in the mutant *monopteros* (*mp*) prevents root formation[30,31]. Identical effects are seen in the *bodenlos* (*bdl*) mutant, in which aberrant AUX/IAA stabilization, due to a mutation in its degron, renders the protein resistant to degradation causing *iaa/bdl* gain-of-function mutants to die during embryogenesis[31,32]. Concomitantly, genetic experiments have shown that reducing the number of functional TIR1/AFBs in plants leads to a variety of auxin-related growth defects, and increased resistance to exogenous auxin, due to compromised AUX/IAA ubiquitylation and turnover[33].

Biochemical and structural analyses in the last two decades have revolutionized our understanding of the mechanisms of auxin sensing and signal transduction. Degron-carrying AUX/IAAs and TIR1/AFB1–5 form an auxin co-receptor system, where auxin occupies a binding pocket in TIR1 just underneath the AUX/IAA degron[23]. Auxin-binding kinetics of the receptor are mainly determined by the specific AUX/IAA binding to TIR1[24]. Hence, different combinations of TIR1/AFBs and AUX/IAAs have different auxin-sensing properties, becoming a versatile co-receptor system for tracing fluctuating intracellular auxin concentrations[24]. While the degron is absolutely necessary for AUX/IAA recruitment and degradation, it does not explain all auxin-binding properties of a TIR1·AUX/IAA receptor pair[24]. Flexible regions outside the primary degron, decorated with specific lysine residues that undergo ubiquitylation in vitro[34], contribute to differential co-receptor assembly[24], AUX/IAA destabilization[35,36], as well as basal protein accumulation[37].

The dynamic range of auxin sensitivity in plant cells, and by default plant growth and development, rely on efficient AUX/IAA processing by the UPS. Particularly in view of the close to 30 AUX/IAA family members, the mechanistic details of this process still remain to be fully understood. Despite their ubiquitous role in signal transduction, research on their singularity and their distinct contribution on auxin sensing, is still in its infancy. At the structural level, it is of outmost relevance to unveil spatial and structural constraints for TIR1·AUX/IAA auxin co-receptor formation. Despite the fact a well-resolved ASK1·TIR1·auxin (IAA)·IAA7 degron crystal structure is available[23], we lack information on how a full AUX/IAA is positioned on TIR1. Thus, there is knowledge gap on whether additional structural features in TIR1 or AUX/IAAs might restrict or facilitate receptor assembly, auxin binding, and AUX/IAA ubiquitylation and degradation.

Here, we study the structural properties of full TIR1·AUX/IAA auxin co-receptor systems, and report on the influence of IDRs in two representative AUX/IAA family members, IAA7 and IAA12 on TIR1·AUX/IAA interactions. Our data demonstrates how an extended AUX/IAA fold promotes recruitment by TIR1, by offering restrained conformational plasticity for correct positioning on TIR1. We also offer a model of how a potential allosteric effect, that fine-tunes TIR1·AUX/IAA interactions, influences AUX/IAA-regulated gene expression.

## Results

**AUX/IAAs exhibit intrinsic structural disorder.** Regions flanking the canonical GWPPVR degron motif influence AUX/IAA protein recruitment by the SCF$^{TIR1}$, impact auxin binding, and AUX/IAA degradation[24,34–37]. A broader sequence context of the AUX/IAA degron might be crucial for the adequate regulation of AUX/IAA processing and turnover, including post-translational modifications (e.g., ubiquitylation), protein–protein interactions and protein–ligand interactions[24,35,36]. To evaluate whether structural flexibility is a common feature among AUX/IAAs, we predicted global structural disorder along the sequences of the 29 *Arabidopsis thaliana* AUX/IAAs in silico (IUPred2A) (Fig. 1a,

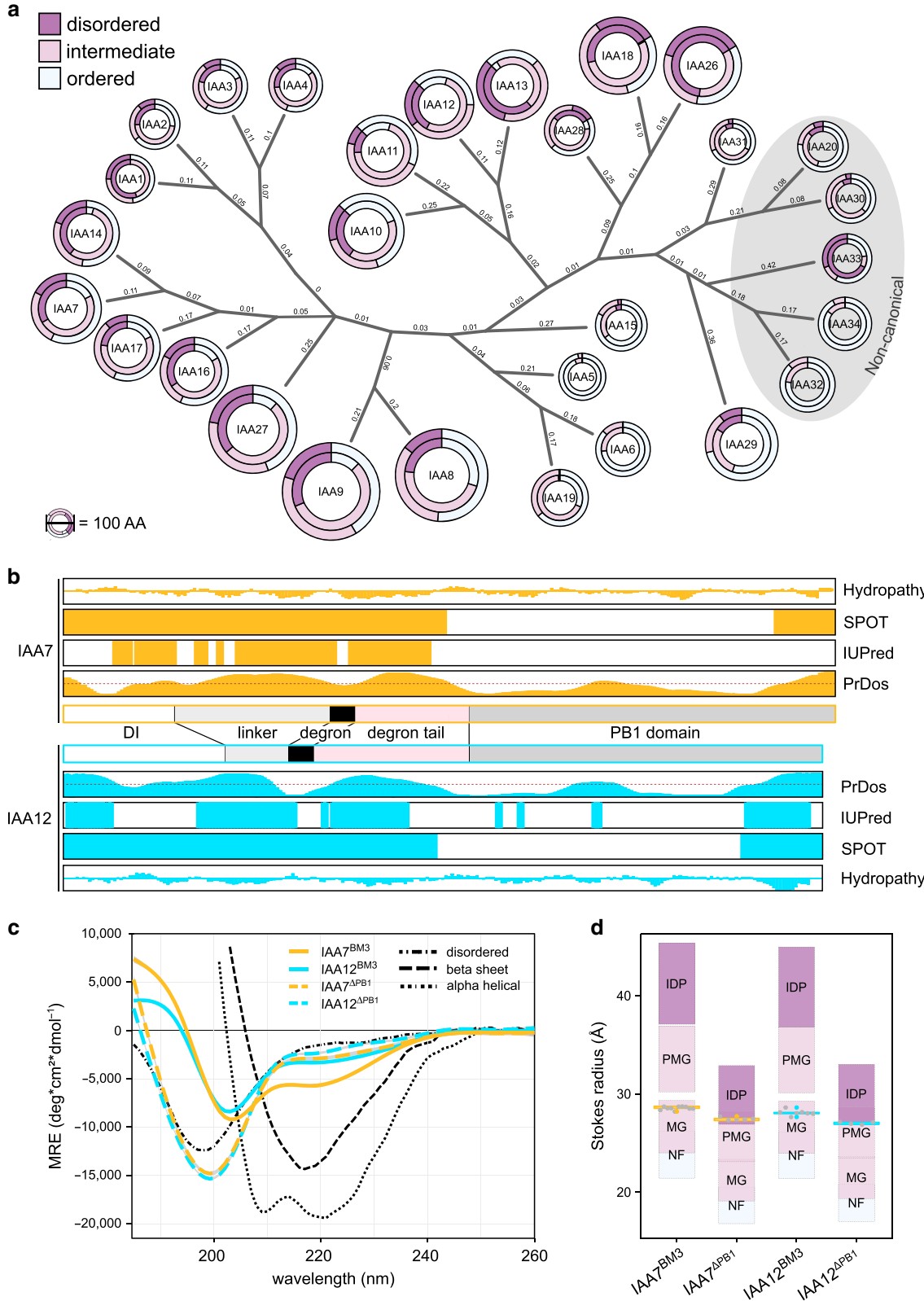

Supplementary Fig. 1, and Supplementary Data 1). We scored the probability of disorder for every amino acid residue in a context-dependent manner[38], and particularly focused on sequences flanking the well-structured PB1 domain (Fig. 1a and Supplementary Fig. 1). We defined scores for disorder probability as high (disordered, >0.6), intermediate (0.4–0.6), or low (ordered, <0.4). AUX/IAA sister pairs arrange in subclades with high sequence similarity, and almost all subclades contain IDRs distributed along their N-terminal halves (NTDs), and much less so, towards the end of their C-terminal PB1 domains (Fig. 1a and Supplementary Fig. 1). The lengths of the AUX/IAAs do not correlate with an enrichment of disorder segments because IAA1-4 or IAA28 (average length below 200 aa) exhibit features of disorder, while similarly small AUX/IAAs (e.g., IAA6, IAA15, IAA19, IAA32, or

**Fig. 1 AUX/IAA proteins are intrinsically disordered outside the PB1 domain. a** Simplified phylogenetic tree (with average branch length depicted) of 29 *Arabidopsis thaliana* AUX/IAAs showing their sequence composition based on IUPred2A prediction for disorder (score classification: disorder (dark lilac): >0.6; intermediate (light lilac): 0.4–0.6; ordered (white): <0.4). Outer circles correspond to full length proteins, inner circles represent disorder prediction excluding the PB1 domain (scale shows width per 100 AA). **b** In silico prediction maps of disorder along the IAA7 (orange) and IAA12 (aquamarine) sequence using SPOT, IUPRED1, and PrDos algorithms. AUX/IAA domain structure (Domain I (DI), a linker, a core degron, a degron tail and the Phox/ Bem1p (PB1) domain) are displayed. Outer plots represent Kyte-Doolittle hydropathy (scale from −4 to +4). Dotted line in PrDos prediction represents a 0.5 threshold. **c** Circular dichroism spectra of IAA7 (orange) and IAA12 (aquamarine) oligomerization-deficient (solid lines) and PB1-less variants (dashed colored lines) show the lack of defined secondary structure elements outside of the PB1 domain. Reference spectra (black dotted lines) are depicted. Ellipticity is calculated as mean residual ellipticity (MRE). Shown is the mean of three independent experiments ($n = 3$). **d** IAA7 (orange) and IAA12 (aquamarine) exhibit an extended fold according to Stokes radii determination via size exclusion chromatography. Theoretical Stokes radii of known folds (lilac color gradient, labeled rectangles): intrinsically disordered protein (IDP, dark lilac), premolten globule (PMG, light lilac), molten globule (MG, light lilac), natively folded (NF, white) plus 10% outer limits, and experimental values (colored box plots with whiskers = ~25% (1.5*IQR) of the data points (gray dots); Outliers shown as colored dots; $n = 4$, 5, 7, and 10 correspondingly for IAA12$^{\Delta PB1}$, IAA7$^{\Delta PB1}$, IAA12$^{BM3}$, IAA7$^{BM3}$).

IAA34) are predicted to be well-structured. With the exception of IAA33, all non-canonical AUX/IAAs, which lack the core degron motif for interaction with TIR1, are rather ordered. IAA33 diverged early during the evolution from the rest of the AUX/ IAAs[28], and it belongs, together with canonical IAA26 and IAA13, to the most disordered family members. Although IAA12 and IAA13 are close ohnologs, IAA13 entails comparatively more disordered segments. IAA7 and IAA12, which are members of a different AUX/IAA subclade[21], appear to have similar bias for IDRs (Fig. 1a and Supplementary Fig. 1).

IAA12 carries a GWPPIG degron that differs from the canonical GWPPVR degron in IAA7, and they equip TIR1·AUX/IAA complexes with distinct auxin binding affinities (TIR1·IAA7: $K_d = $ ~20 nM and TIR1·IAA12: $K_d = $ ~250 nM)[24]. Nevertheless, these differences cannot be solely attributed to the identity of the degron[24]. Therefore, IDRs flanking the degron could probably participate in interactions with TIR1, affecting auxin sensitivities. In order to investigate the distribution of disorder in IAA7 and IAA12 proteins, we performed in silico analyses using multiple disorder prediction algorithms (Fig. 1b). Consistently, all tested algorithms show that most of the disorder segments in IAA7 and IAA12 are located on their NTDs. We also observe an enrichment of hydrophilic residues in these IDRs based on the hydropathy index, indicating that these regions may be solvent exposed (Fig. 1b). Almost 50% of IAA7 and IAA12 amino acid content correspond to disordered regions. In IAA7, but most notably in IAA12, we observe a predominant "order-dip" corresponding to the core degron (Fig. 1b).

In order to obtain hints for IDR presence in IAA7 and IAA12 in solution, we used recombinantly expressed proteins, and further analyzed their secondary structure and overall shape via CD spectroscopy and size exclusion chromatography (Fig. 1c, d and Supplementary Figs. 2 and 3). We looked into a functionally relevant transient AUX/IAA fold, while considering different protein conformational classes (Fig. 1c). We included oligomerization-deficient variants IAA7$^{BM3}$, IAA12$^{BM3}$ (ref. [39]), and also IAA7 and IAA12 truncated variants lacking the compactly folded PB1 domain, IAA7$^{\Delta PB1}$ and IAA12$^{\Delta PB1}$. Both IAA7$^{BM3}$ and IAA12$^{BM3}$ exhibit a rather complex mix of secondary structure elements characteristic of premolten globule–like proteins, displaying a minimum at ~205 nm, and a shoulder near 220 nm in CD spectra[40]. CD spectra of IAA7$^{\Delta PB1}$ and IAA12$^{\Delta PB1}$ shifted toward a shorter wavelength with a minimum at just below 200 nm, which is characteristic for random coil proteins (Fig. 1c and Supplementary Fig. 2). We also measured the Stokes radii ($R_S$) for IAA7$^{BM3}$, IAA12$^{BM3}$ together with the theoretical values of IAA7 and IAA12 displaying specific folds, native fold (NF), molten globular (MG), premolten globule (PMG), and unfolded (IDP) (Fig. 1d). Since all measured Stokes radii are larger than the ones expected for their respective natively folded proteins, we conclude that IAA7$^{BM3}$ and IAA12$^{BM3}$ adopt extended

structures mainly due to large proportions of intrinsically disordered segments outside of the PB1 domain.

**Intrinsic disorder impacts auxin-driven receptor association.** As IAA7 and IAA12 have distinct and contrasting TIR1-interaction properties, we reasoned generating IAA7 and IAA12 chimeric proteins could enable to pinpoint the contribution of IDRs flanking the degron to auxin-dependent TIR1·AUX/IAA associations. IAA7 and IAA12, as well as their sister proteins IAA14 and IAA13, respectively, exhibit differences in their disordered degron tail length and charge distribution (Supplementary Fig. 4). While IAA7 and IAA14 have in average a short degron tail (<30 aa), IAA12 and IAA13 have a longer degron tail (44 aa) linking the degron to the PB1 oligomerization domain (Supplementary Fig. 4). We defined five different modules flanked by motifs conserved throughout the AUX/IAA family: DI (N-terminus including KR motif), core degron (VGWPP-[VI]-[RG]-x(2)-R), the PB1 domain, and two variable IDRs connecting either the DI and degron (linker), or the degron and PB1 domain (degron tail) (Fig. 2a and Supplementary Fig. 4). We exchanged the modules between IAA7 and IAA12 and used the resulting seamless chimeras in the yeast two-hybrid system (Y2H) to assess their respective ability to interact with TIR1 (Fig. 2a, Supplementary Fig. 4, and Supplementary Data 1). As previously reported, we find native IAA7, denoted here 7-7-7-7-7, interacts with TIR1 in an auxin-dependent manner more strongly than native IAA12 (12-12-12-12-12). Mimicking degron mutants iaa7/axr2-1 (P87S) or iaa12/bdl (P74S) in the IAA7 or IAA12 chimeras (7-7-7m-7-7, 12-12-12m-12-12) abolishes, expectedly, their association with TIR1 (Fig. 2a). Exchanging the disordered degron tail of IAA7 for the one in IAA12 in the IAA(7-7-7-12-7) chimera does not affect interaction with TIR1. A IAA(12-12-12-7-12) chimera, however, associates with TIR1 much more efficiently than wild type IAA (12-12-12-12-12). Similarly, PB1 domain exchanges between IAA7 or IAA12 positively affect the ability of IAA(12-12-12-12-7) chimera to interact with TIR1. To investigate the interdependency of the degron tail and the PB1 domain, we exchanged the flexible degron tail of IAA12 together with its corresponding PB1 domain, and fused them to IAA7 (IAA(7-7-7-12-12)). In this case, TIR1·IAA(7-7-7-12-12) interaction is greatly affected, while TIR1·IAA(12-12-12-7-7) interaction, although weak, remains stronger than TIR1·IAA(12-12-12-12-12) association. Of note, independently of the specific core degron sequence, GWPPVR in IAA7 or GWPPIG in IAA12, the IAA7 degron tail, and PB1 combo of IAA7 favors auxin-dependent TIR1·AUX/IAA chimera interactions. Furthermore, alterations in the IAA7 domain structure interferes with its degradation. (Supplementary Fig. 5). Taken together, auxin-dependent and -independent interactions are influenced by the degron tail, as well as the PB1 domain, and these regions may act in concert.

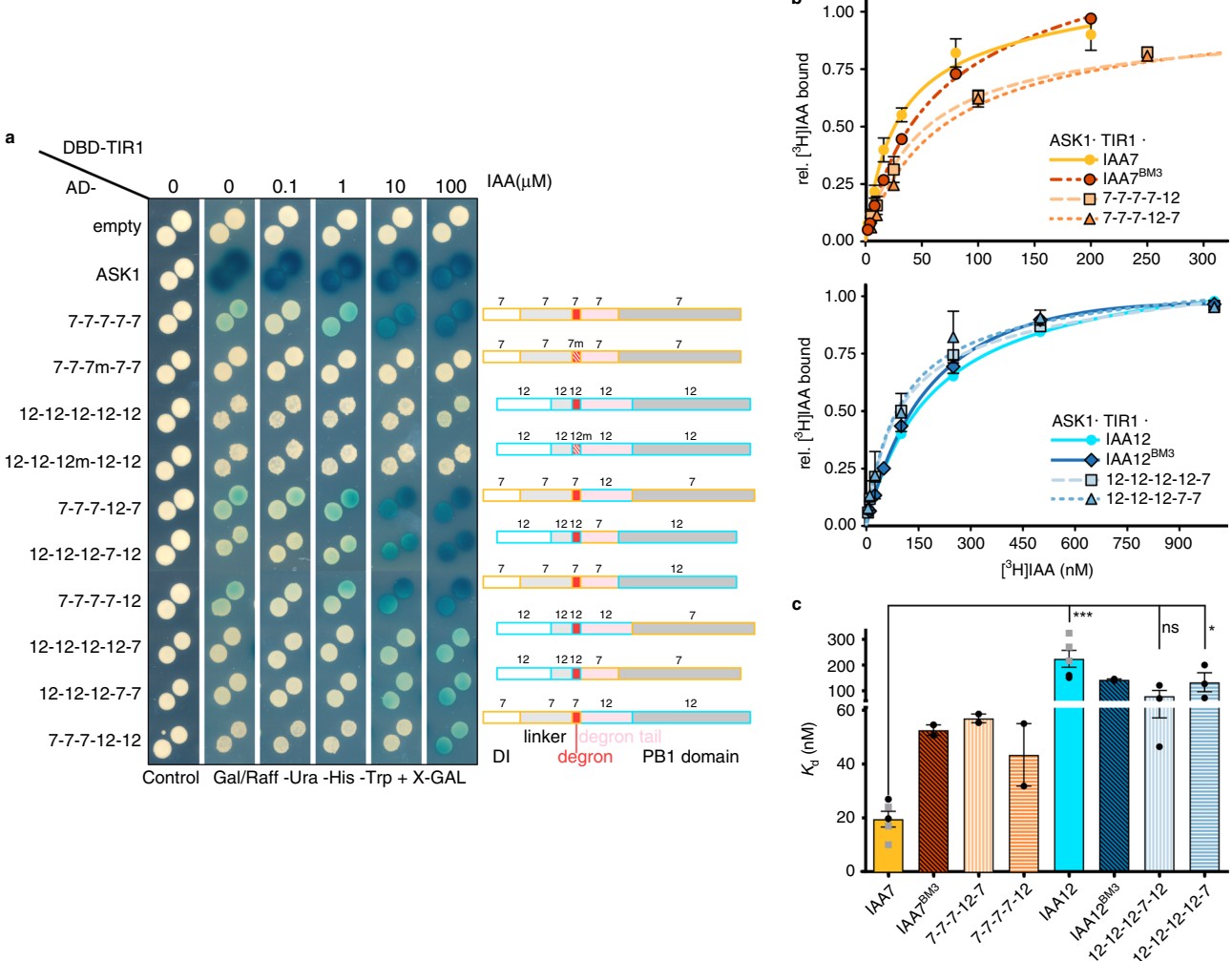

**Fig. 2 Auxin-dependent TIR1·IAA7 and TIR1·IAA12 interactions rely on the interplay between the degron, degron tail and the PB1 domain. a** Y2H interaction matrix (left) for TIR1 with ASK1, and ten chimeric proteins built by fusing IAA7 and IAA12 segments flanked by conserved motifs throughout the AUX/IAA family. Yeast diploids containing LexA DBD-TIR1 and AD-AUX/IAA chimeras were spotted to selective medium with increasing IAA concentrations, and β-galactosidase reporter expression indicated TIR1·AUX/IAA interactions. AD-empty vector as negative control. Domain organization and composition of seamless chimeric IAA7 (orange) and IAA12 (aquamarine) constructs depicted in boxes (right) with DI (white) (till KR motif), linker (light gray), core degron (red), degron tail (light pink), and PB1 domain (dark gray). **b** Saturation binding assays using [³H]IAA and recombinant ASK1·TIR1·IAA7 (shades of orange) or ASK1·TIR1·IAA12 (shades of blue) ternary complexes. TIR1·IAA7 complex exhibits a high affinity ($K_d$ = ~20 nM) for auxin, whereas IAA12-containing co-receptor complexes provide tenfold lower affinity for auxin ($K_d$ = ~200 nM). Oligomerization-deficient IAA7$^{BM3}$ and IAA12$^{BM3}$ variants, and chimeric AUX/IAA proteins in complex with ASK1·TIR1 distinctly affect auxin bind capabilities of a co-receptor system. Shown are saturation binding curves for each co-receptor pair as relative [³H]IAA binding normalized to the highest value of each curve. Each point reflects means of 2–3 independent experiments ($n$ = 3 for 12-12-12-7-12 and 12-12-12-12-7; otherwise $n$ = 2), each of them comprising of technical triplicates, and depicted as means ± SEM. **c** Comparison of dissociation constants ($K_d$) obtained in saturation-binding experiments for each ASK1·TIR1·AUX/IAA ternary complex. Shown are mean values from our experiments (black dots) combined with published[24] $K_d$ values for IAA7 and IAA12 (gray squares) $K_d$ values depicted as means ± SEM ($n$ = 2 + 3 for IAA7 and IAA12; $n$ = 3 for 12-12-12-7-12 and 12-12-12-12-7: otherwise $n$ = 2). For $n$ ≥ 3, significant differences are indicated (Dunnett's multiple comparison test, IAA7 as reference; *$p$ < 0.0322, ***$p$ < 0.0002).

In order to address whether accessibility of IDRs and the PB1 domain in AUX/IAAs affect the outcome of TIR1·AUX/IAA interactions, that is auxin sensing, we performed in vitro radio-oligand binding assays with TIR1, IAA7, and IAA12 wild type, chimeric, as well as IAA7$^{BM3}$ and IAA12$^{BM3}$ mutant proteins. Thereby, we also indirectly assayed whether AUX/IAA homo- and hetero-dimers, through the PB1 domain, might impinge on auxin binding. While PB1-compromised IAA7 (IAA7$^{BM3}$) together with TIR1 shows diminished auxin binding affinity, IAA12$^{BM3}$ does not interfere with the auxin binding properties of the receptor complex (Fig. 2c). We observed the general trend of reduced auxin binding affinities when altering IAA7 in TIR1·IAA7$^{BM3}$

($K_d$ = ~53 ± 2 nM), TIR1·IAA (7-7-7-12-7) ($K_d$ = ~57 ± 2 nM), and TIR1·IAA (7-7-7-7-12) ($K_d$ = ~43 ± 12 nM) complexes in comparison to TIR1·IAA(7-7-7-7-7) ($K_d$ = ~20 ± 3 nM) ($n$ = 2–5; ± indicates SEM). We, however, measured relatively similar auxin affinities of TIR1·IAA12$^{BM3}$ ($K_d$ = ~143 ± 3 nM), TIR1·IAA (12-12-12-7-12) ($K_d$ = ~79 ± 22 nM), TIR1·IAA (12-12-12-12-7) ($K_d$ = ~133 ± 37 nM), and TIR1·IAA12 ($K_d$ = ~226 ± 34 nM); ($n$ = 2–5; ± indicates SEM) (Fig. 2b, c and Supplementary Fig. 6). The decrease in the auxin binding affinity of TIR1·IAA7$^{BM3}$ and TIR1·IAA (7-7-7-7-12) hints to a positive effect of the IAA7 PB1 domain to auxin sensing (Fig. 2c). The degron tail, as well as the PB1 domain of IAA12 in the IAA7 context, reduce the

auxin binding affinity by around fourfold (Fig. 2b, c). Conversely, we did not trace a significant effect of individual IAA7 modules when inserted in the IAA12 context (Fig. 2a). This is consistent with our Y2H interaction data, evidencing the specific interdependency of degron tails and their corresponding PB1 domains. It further points to additive and separate effects of each intrinsically disordered degron tail and the PB1 domain on auxin-independent and auxin-dependent TIR1 interaction.

**IDRs in AUX/IAAs facilitate their ubiquitylation**. To next examine the contribution of the AUX/IAA IDRs to their ubiquitylation by the SCF$^{TIR1}$ complex, we recapitulated auxin-triggered and SCF$^{TIR1}$-dependent IAA7 and IAA12 ubiquitylation in vitro (IVU)[34]. We followed IAA7 and IAA12 ubiquitylation over time using IAA concentrations in the range of their auxin binding affinity ($K_d$) of TIR1·IAA7 and TIR1·IAA12 complexes (i.e., ~20 nM to ~200 nM) and beyond (Figs. 2c and 3, and Supplementary Fig. 7). In our IVUs, AUX/IAA ubiquitylation is detectable as early as 10 min after incubation, and accelerated in an auxin-dependent manner. In the absence of auxin, IAA12~ubiquitin conjugates are less abundant than IAA7~ubiquitin conjugates (Fig. 3a and Supplementary Fig. 7). Differences between IAA7 and IAA12 ubiquitylation are prominent at shorter incubation times, and especially at concentrations below 150 nM (Fig. 3). We figured IAA7 and IAA12 ubiquitylation occurs rapidly, and differences in their ubiquitylation dynamics depend on the auxin binding affinities of their respective TIR1·AUX/IAA receptor complex. This is possibly the result of an increased dwell-time of the AUX/IAA on TIR1, which facilitates efficient ubiquitin transfer to lysines.

Putative ubiquitin acceptor Lys residues along the IAA7 and IAA12 sequences are enriched in the degron tail of IAA12, and the linker of IAA7, both of which appear to lack a three-dimensional (3D) structure (Fig. 3b). We aimed at gaining experimental evidence of IAA7 and IAA12 ubiquitylation sites, after IVU reactions, tryptic digest and LC/MS analysis. We were able to map only few specific lysine residues on IAA7 and IAA12, which are differently distributed along their sequence (Fig. 3b and Supplementary Data 2). Although IAA7 and IAA12 contain 24 and 18 lysines, respectively, only 3 and 6 of them were ubiquitylated. While we observe only few ubiquitylated lysine residues at the AUX/IAA N-terminus, most of the mapped ubiquitylation sites are located in the region downstream of the degron, either in the PB1 domain in IAA7, or the degron tail in IAA12. Even though 4 lysines are conserved in the PB1 domain of IAA7 and IAA12, only the non-conserved residues appear to be ubiquitylated in IAA7. The flexible degron tail of IAA7 did not get ubiquitylated, whereas 4 out of 7 lysines in the slightly longer disordered IAA12 degron tail could be mapped as ubiquitylation sites (Fig. 3b and Supplementary Data 2).

To further investigate whether the apparent structural divergence of IAA7 and IAA12 imposes restrictions to lysine access for ubiquitylation, we used chimeric IAA7 and IAA12 proteins in our IVU assay (Fig. 3c). As we aimed at visualizing absolute differences in ubiquitin conjugation, we traced auxin-dependent ubiquitin conjugation of chimeric AUX/IAAs at a fixed IAA concentration of 1 μM after 1 h IVU reaction. Exchanging the degron tails or the PB1 domains between IAA7 and IAA12 leads to differences in ubiquitylation profiles of chimeric proteins compared to their wild type counterparts. This happens as we either added or subtracted regions that contain the ubiquitin acceptor sites in the IAA7 and IAA12 chimeric proteins (Fig. 3c and Supplementary Fig. 8). For instance, we detect an increase of ubiquitin conjugates on IAA(7-7-7-12-7), which gains ubiquitylation sites due to the exchange of the IAA7 degron tail. Deleting the AUX/IAA degron tail or the PB1 domain in the

chimeric proteins results in an overall reduction of ubiquitin conjugates on targets. Versions of IAA7 or IAA12 missing a degron tail and containing the PB1 domain of IAA12, IAA(7-7-7-Δ-12) and IAA(12-12-12-Δ-12), do not undergo auxin-triggered ubiquitylation (Fig. 3c and Supplementary Fig. 8). Similarly, AUX/IAA versions containing the IAA7 degron, but lacking a PB1 domain (IAA(7-7-7-Δ), and IAA(12-12-12-7-Δ)) are not conjugated by ubiquitin, probably due to the loss of the mapped ubiquitin acceptor sites (Fig. 3b). Our IVU assays on AUX/IAA chimeras validate our findings showing that the IAA7 PB1 domain or the flexible IAA12 degron tail carry propitious ubiquitylation sites. Thus, we postulate AUX/IAA ubiquitylation favorably occurs in exposed regions in IAA7 and IAA12, when they are recruited by TIR1.

**Degron-flanking regions tailor TIR1·AUX/IAA ensembles.** Owing to the relative lack of a stable 3D conformation, IDPs or proteins enriched in IDRs, such as AUX/IAAs, represent a challenge for structural biology studies. During interactions with target proteins, IDPs, particularly their IDRs, may undergo conformational changes that cannot be traced easily, or captured while happening[41,42]. Although the ASK1·TIR1·auxin (IAA)·IAA7 degron crystal structure enlightened us on how auxin is perceived, we lack information on the contribution of regions flanking the AUX/IAA degron on auxin binding. Without being able to structurally resolve intrinsically disordered degron-flanking regions, we are hindered in our understanding of how AUX/IAAs are actually positioned on TIR1. This has evidently far-reaching implications on SCF$^{TIR1}$ E3 ubiquitin ligase activity and ubiquitin transfer to AUX/IAAs by an E2 ubiquitin conjugating enzyme.

To elucidate the driving factors for ASK1·TIR1·AUX/IAA complex assembly, and to unveil how IDRs in AUX/IAAs influence positioning on TIR1, we pursued a structural proteomics approach using chemical cross-linking coupled to mass spectrometric analyses (XL-MS) (Fig. 4a). We assembled ASK1·TIR1·AUX/IAA complexes containing either IAA7$^{BM3}$ or IAA12$^{BM3}$ proteins in the absence or presence of auxin, and added the MS-cleavable crosslinker disuccinimidyl dibutyric urea (DSBU). Reaction products were processed for mass spectrometric analysis, which utilizes the characteristic fragmentation of DSBU to identify crosslinked residues within the AUX/IAAs and the ASK1·TIR1·AUX/IAA complex[43–45]. Our data shows multiple intra- and inter-molecular crosslinks (XLs) for ASK1·TIR1 and IAA7$^{BM3}$ or IAA12$^{BM3}$ proteins when auxin was included (Fig. 4b, c, Supplementary Fig. 9, Supplementary Data 3 and 4). In the absence of auxin, we observe only few inter-protein and similar intra-protein XLs when compared to auxin-containing samples (Fig. 4c, Supplementary Fig. 10, and Supplementary Data 3 and 4). In the presence of auxin, we identify two distinct clusters in TIR1 harboring crosslinker-reactive amino acid side chains with IAA7 and IAA12 (Fig. 4b and Supplementary Fig. 9). Cluster 1 comprises amino acid residues in LRR4–7 (140–229 aa), while cluster 2 consists of residues toward the TIR1 C-terminus located in LRR17–18 (485–529 aa). The location of the clusters on two opposing surfaces of TIR1 suggests a rather extended fold of the AUX/IAA protein when bound to TIR1 (Fig. 4b). The crosslinked residues along the sequences of ASK1·TIR1·IAA7$^{BM3}$ or ASK1·TIR1·IAA12$^{BM3}$ show an enrichment of highly variable intramolecular XLs within the AUX/IAAs (Fig. 4c). A low number of intra-protein XLs along the TIR1 sequence were detected as a consequence of its rigid solenoid fold, which is in agreement with the ASK1·TIR1 crystal structure (PDB: 2P1Q [http://www.rcsb.org/structure/2P1Q][23]). Inter-protein XLs indicate that the crosslinker-reactive clusters in TIR1 mainly connect with only a specific subset of AUX/IAA residues (Fig. 4b, c). Multiple IAA7

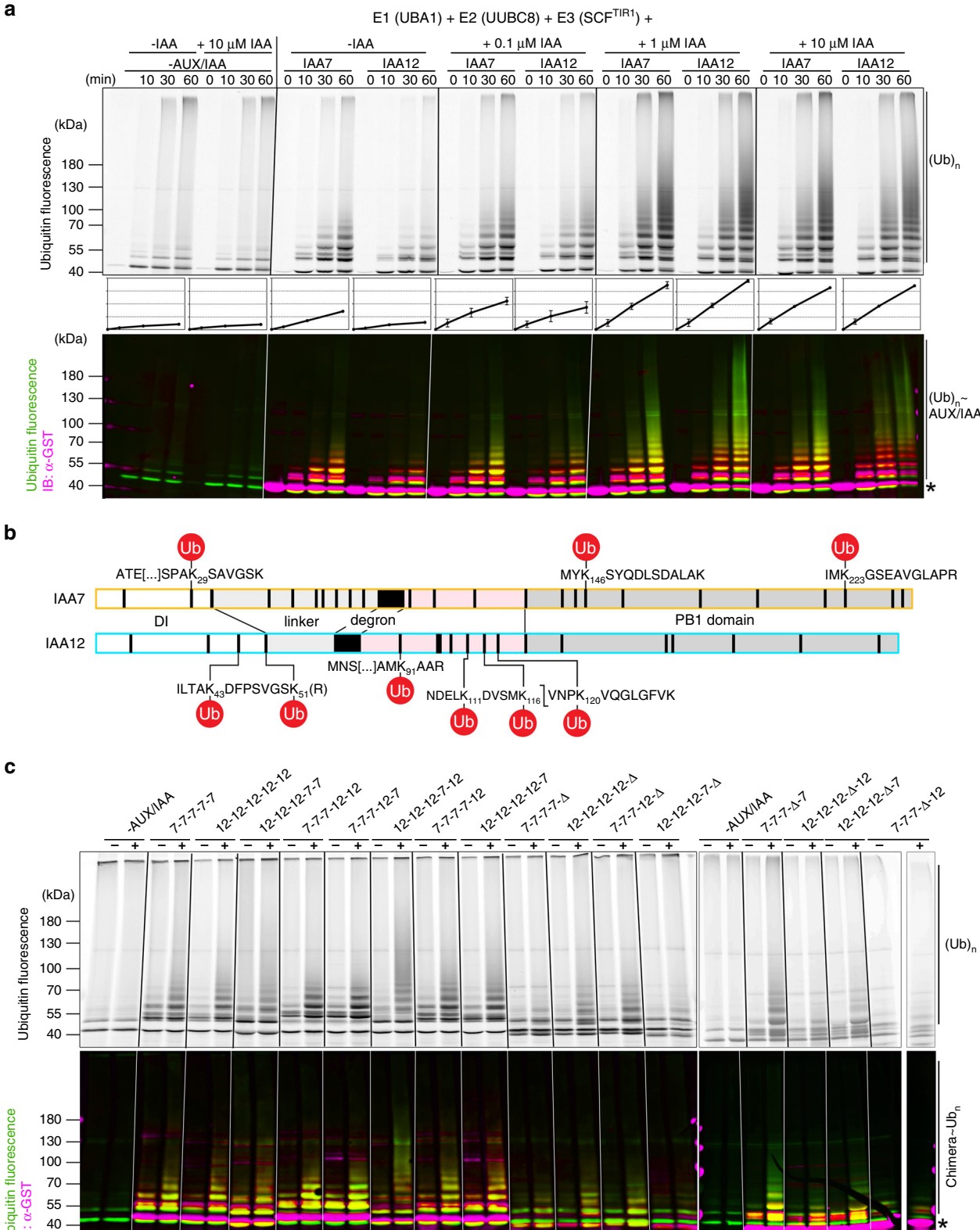

**Fig. 3 Auxin-driven and SCF<sup>TIR1</sup>-dependent ubiquitylation of IAA7 and IAA12 display distinct dynamics. a** IVU assays with recombinant GST-IAA7 or GST-IAA12, E1 (*At*UBA1), E2 (*At*UBC8), reconstituted SCF<sup>TIR1</sup> (*At*SKP1·TIR1, *Hs*Cul1 and *Mm*RBX1), fluorescein-labeled ubiquitin (Ub) and IAA (auxin). IAA7 and IAA12 ubiquitylation is auxin-driven and time-dependent. Ubiquitylation was monitored using the ubiquitin fluorescent signal (green), and anti-GST/Alexa Fluor 647-conjugated antibodies for detection of GST-AUX/IAAs (magenta). ImageQuantTL software was used for quantification (middle; means ± SEM, $n = 3$), and generation of merged image (bottom; overlapping Ub and GST signal: yellow). **b** IAA7 and IAA12 IVU samples were analyzed via LC-MS, and putative ubiquitylation sites, detected by the diGly (or LRGG) Ub remnant after tryptic digest, were mapped relative to the domain structure. IAA12 Ub sites agglomerate in the region upstream of the degron (white) and the degron tail (light pink). **c** Ubiquitin conjugation on chimeric IAA7 and IAA12 (colors as in **a**) proteins in the presence or absence of 1 μM IAA. IVU reaction time 1 h. (\*) Asterisks depict unmodified AUX/IAAs.

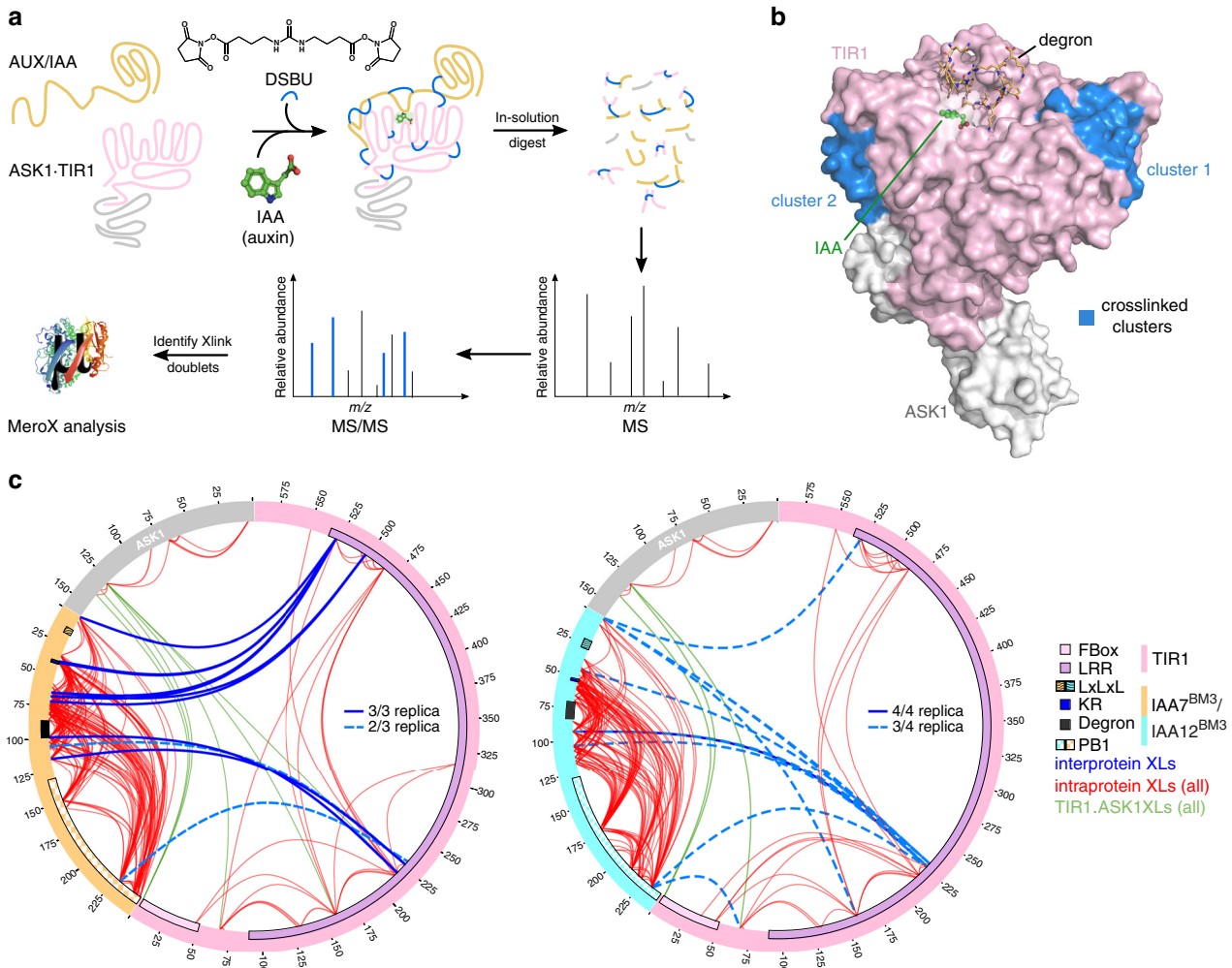

**Fig. 4 Structural proteomics using an MS-cleavable crosslinker reveals TIR1·IAA7 and TIR1·IAA12 interaction interfaces. a** Workflow for the cross-linking coupled to mass spectrometry (XL-MS) approach. Recombinant oligomerization-deficient IAA7 (orange) and IAA12 (aquamarine) proteins, and ASK1·TIR1 (gray and light pink, respectively) were incubated with the DSBU crosslinker, and samples were analyzed using LC/MS[2]. Crosslinked peptides were identified using the MeroX software. **b** Interaction interfaces (blue) on TIR1 converge in two distinct patches around LRR4–7 (cluster 1) or LRR17–18 (cluster 2) revealing AUX/IAAs adopt an extended fold when in complex with TIR1. **c** Circular depiction of inter-protein (TIR1·AUX/IAAs) (blue) and intra-protein (red) crosslinks (XLs) along IAA7 (orange), IAA12 (aquamarine), TIR1 (light pink) and ASK1 (gray) protein sequence. XLs were identified in at least 2/3 or 3/4 independent experiments (dashed: 2/3 and 3/4; solid lines: 3/3 and 4/4). Specific XLs within TIR1 and between TIR1 and ASK1 (green) are in agreement with the crystal structure (PDB: 2P1Q [http://www.rcsb.org/structure/2P1Q]). Known motifs and protein domains are displayed.

residues upstream of the core degron, including the KR motif, preferably crosslinked to TIR1 cluster 2. While residues down-stream of the core degron, including the PB1 domain, positioned towards TIR1 cluster 1 (Fig. 4c). Interestingly, degron-neighboring residues, populating the most stable part of TIR1·AUX/IAA complexes, are highly represented in the XL data sets (Supplementary Data 3 and 4). IAA12 is similarly positioned on TIR1, but exhibits even higher flexibility given the more diverse distribution of inter-protein XLs (Fig. 4c). This is also supported by the fact that we detect many more assemblies for ASK1·TIR1·IAA12[BM3] across replicates, than for the ASK1·TIR1·IAA7[BM3] complex (Fig. 4c). In conclusion, our structural proteomics approach confirms AUX/IAAs IAA7 and IAA12 exhibit flexible conforma-tions in solution (intra-protein XLs), and adopt an extended fold when bound to TIR1.

As we gained a better understanding of the extended fold of IAA7 and IAA12 on TIR1, we wondered whether intrinsic disordered stretches flanking the degron might help to coordinate positioning of the AUX/IAA PB1 domain for ubiquitin transfer. An extended AUX/IAA configuration on TIR1 would be

particularly relevant for allowing K146 and K223 in the PB1 domain of IAA7 to be readily available for ubiquitylation. In the case of IAA12, an assertive extension of the degron tail would expose K91, K111, K116, and K120 for ubiquitin attachment (Fig. 3b).

**Conformational heterogeneity steers AUX/IAA interactions.** To further investigate how intrinsic disorder in IAA7 and IAA12 influences their positioning on ASK1·TIR1, we combined our XL information with a molecular docking strategy (Fig. 5 and Supplementary Figs. 11 and 12). For that, we used available structures for the PB1 domains of AUX/IAAs and ARFs[39,46–48]. We docked homology-modeled PB1 domains of *Arabidopsis* IAA7 and IAA12 to the ASK1·TIR1 complex, applying distance restraints based on the XL data using HADDOCK (Supplementary Fig. 12 and Supplementary Tables 1 and 2). We also added an additional distance restraint reflecting the possible conformational space covered by the respective degron tails. We visualized the impact of the different restraints on the possible interaction interface of ASK1·TIR1·IAA7[PB1] and ASK1·TIR1·IAA12[PB1] by DisVis[49]

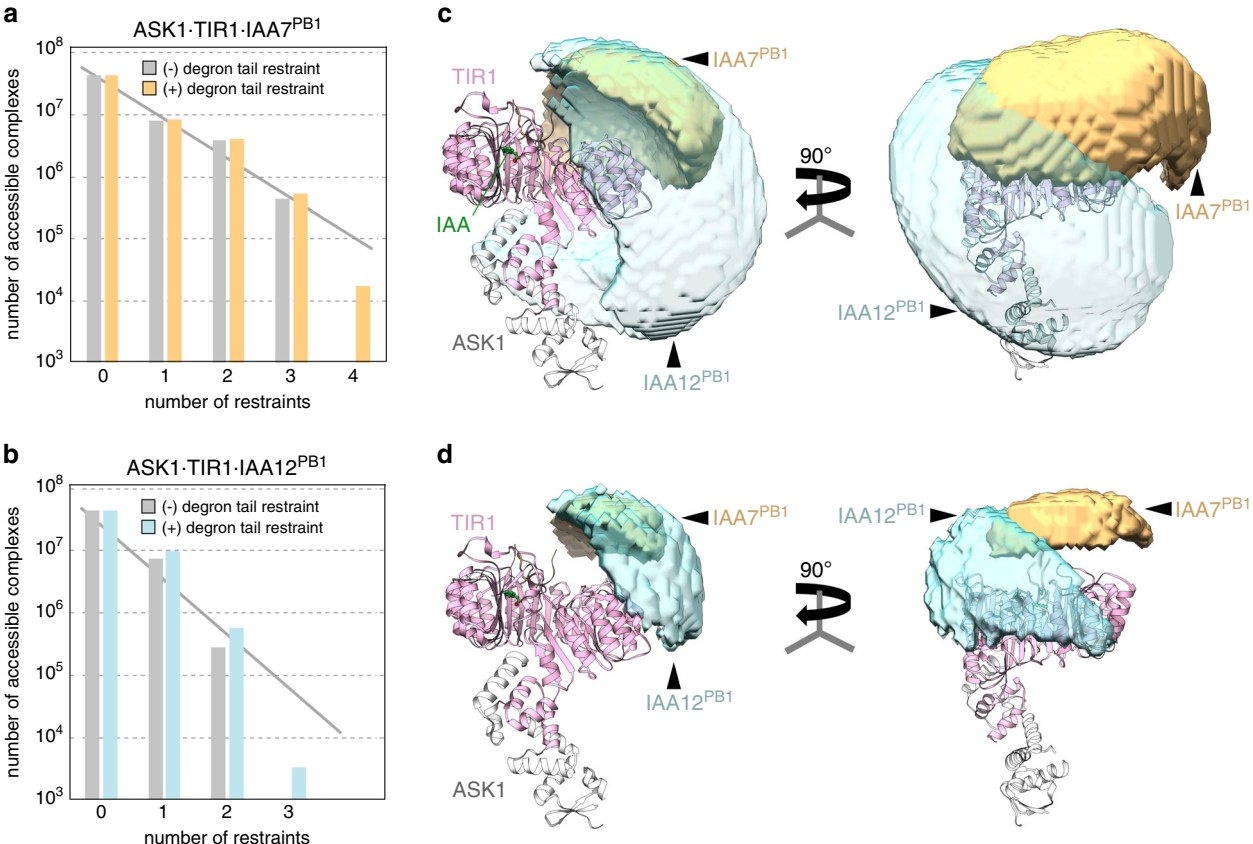

**Fig. 5 XL-based docking substantiates the function of the AUX/IAA disordered degron tail positioning the PB1 domain on TIR1.** Conformational space available for docking the PB1 domains of IAA7 (light orange) and IAA12 (aquamarine) on ASK1·TIR1 (gray, light pink) analyzed by DisVis based on the XL data restraints. **a**, **b** Including the length of the disordered degron tail of IAA7 (36 aa) or IAA12 (49 aa) as an additional restraint (colored bars) conspicuously reduces the conformational space, and the number of accessible TIR1·AUX/IAA$^{PB1}$ complexes. **c**, **d** Visualization of the possible conformational space occupied by the PB1 domain on the ASK1·TIR1 protein complex without (**c**) or with (**d**) the degron tail length as distance restraint.

(Fig. 5). Evidently, by incorporating more distance restraints, we limit the number of ASK1·TIR1·AUX/IAA$^{PB1}$ protein complexes, therefore reducing their explored interaction space (Fig. 5).

Intriguingly, the relationship between the accessible complexes vs. the number of restraints applied does not reveal a linear behavior, but shows a sharp drop when the degron tail restraint is added to all XL-based restraints (Fig. 5a, b). Comparing the groups of water-refined HADDOCK models leads to similar observations, and the best scoring groups are only sampled incorporating the degron tail restraint (Supplementary Tables 1 and 2). This indicates the disordered degron tail in AUX/IAAs restricts the conformational space explored by the PB1 domain on TIR1 (Supplementary Table 2 and Supplementary Fig. 12). The reduction of accessible ASK1·TIR1·IAA7$^{PB1}$ and ASK1·-TIR1·IAA12$^{PB1}$ complexes for docking is also reflected by the decreased space that can be possibly occupied by the PB1 domain (Fig. 5c, d). Overall, XL-based docking of the PB1 domain of IAA12 on the ASK1·TIR1 complex is less-defined, and occupies a distinct conformational space than the ASK1·TIR1·IAA7$^{PB1}$ complex.

In order to refine our docking data and identify the most energetically favored TIR1·AUX/IAA$^{PB1}$ assemblies, we carried out molecular dynamic simulations coupled to free-binding energy calculations by MM/GBSA. We used as a starting structure ($t = 0$) the results from the HADDOCK simulations, including the degron tail restraint (cluster1_1; 2_1 (IAA7 and IAA12); 3_1(IAA12)), and performed 20 ns simulations for each TIR1·IAA7$^{PB1}$ or TIR1·IAA12$^{PB1}$, resulting in stable

complexes (Fig. 6a and Supplementary Fig. 12). We obtained the effective binding free-energy every 1 ps for each simulation, and observed distinct average effective energy ($\Delta G_{eff}$) for the different groups in each system (TIR1·IAAx$^{PB1}$ protein complex). Group 1 for TIR1·IAA7$^{PB1}$ and groups 1 and 3 for TIR1·IAA12$^{PB1}$ turn out to be energetically less favored, while groups 2 in each case show the lowest binding energy. This indicates groups 2 likely depict the most probable ensembles (Fig. 6a). We further carried out per-residue effective energy decomposition analysis (prEFED) followed by validation via computational alanine scanning (CAS) in order to identify relevant residues in groups 2 favoring TIR1·AUX/IAA$^{PB1}$ interactions (Fig. 6b and Supplementary Table 3). We found residues in TIR1 that might engage in polar interactions with the AUX/IAA PB1 domain. D119, D170, V171, S172, H174, H178, S199, R220 along the LRR3–7 in TIR1 likely contribute to stabilization of the TIR1·IAA7$^{PB1}$ complex. Residues H174, H178, S199 also stabilize TIR1·IAA12$^{PB1}$ interactions together with R156, S177, S201, and R205 in TIR1 LRR4–6 (Fig. 6b, c and Supplementary Fig. 13).

**A paradigm for TIR1·auxin·AUX/IAA interactions in vivo.** To next determine whether the in silico identified TIR1 residues contribute to its function, and therefore auxin receptor formation, we first generated mutant proteins and evaluated their interaction potential in Y2H assays (Fig. 7a and Supplementary Fig. 14). Mutations S199A and R220A impair ASK1·TIR1, TIR1·IAA7, as

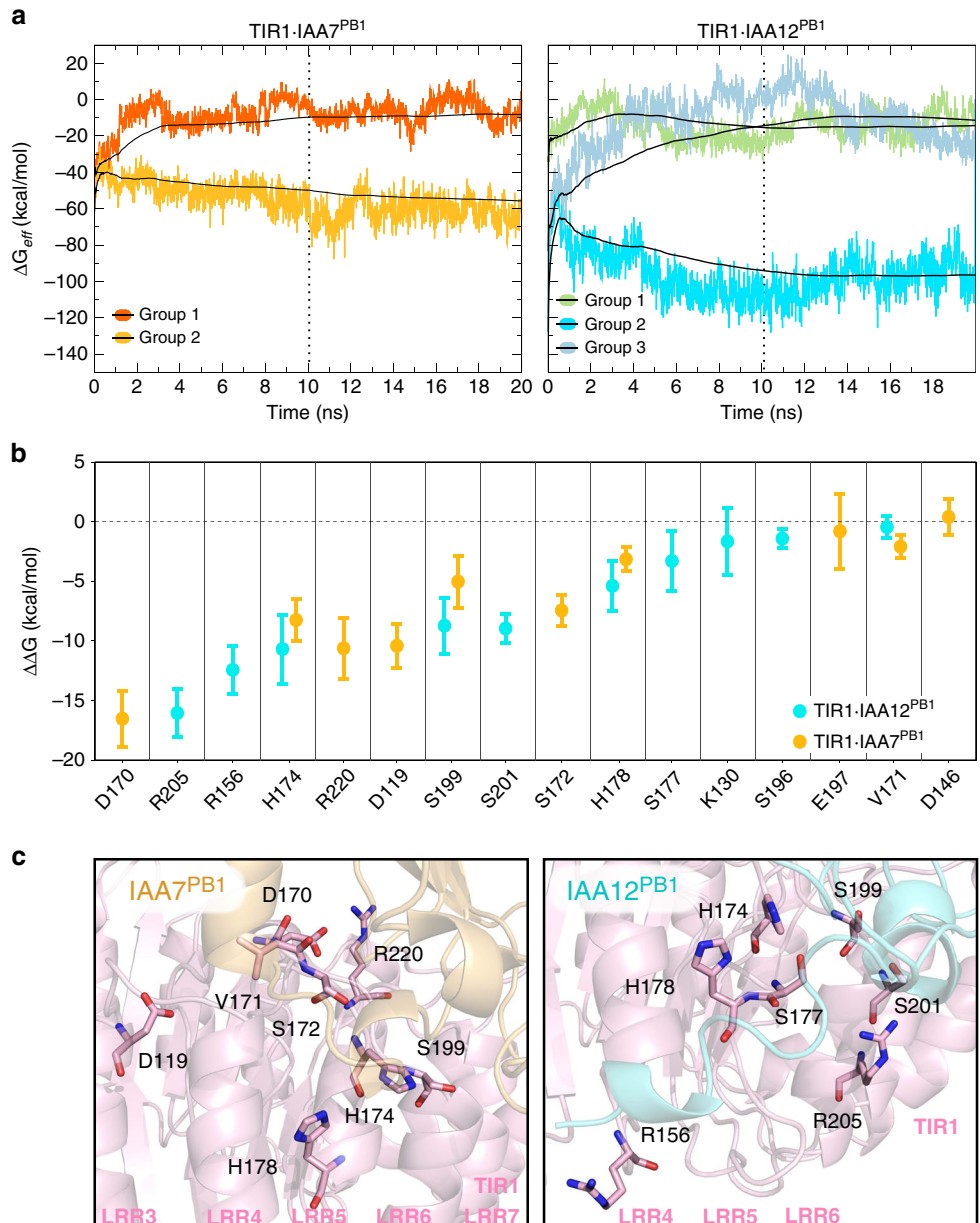

**Fig. 6 Molecular dynamics simulations reveal energetically favorable TIR1·AUX/IAA$^{PB1}$-interacting moieties. a** Time evolution of instantaneous $\Delta G_{eff}$ values over 20 ns identifying stable TIR1·AUX/IAA$^{PB1}$ complexes from HADDOCK best scoring groups (IAA7: cluster1_1; 2_1; and IAA12: cluster1_1; 2_1; 3_1). Black lines indicate the accumulated mean values of $\Delta G_{eff}$ for each trajectory. One low energetic and stable complex (group 2, light orange (IAA7) or aquamarine (IAA12)) was identified for TIR1·IAA7$^{PB1}$ (dark and light orange), and TIR1·IAA12$^{PB1}$ (blues and green) systems. Dotted vertical line at 10 ns indicates the time point of equilibrium used as a reference for subsequent analysis. **b** Energetically relevant TIR1 residues for TIR1·AUX/IAA$^{PB1}$ (TIR1·IAA7$^{PB1}$: orange; TIR1·IAA12$^{PB1}$: aquamarine) complex formation were identified by computational alanine scanning (CAS) using MD trajectories (in **a**) from the equilibration time point onwards (depicted as means ± SEM). **c** Stick representation of CAS-identified residues in TIR1 (light pink) localize to the LRR3–7 forming a polar patch that allows interaction with either IAA7$^{PB1}$ (orange) or IAA12$^{PB1}$ (aquamarine).

well as TIR1·IAA12 interactions. This implies these changes cause a long-range effect on TIR1 activity, and probably its overall conformational stability. Mutations S201A, D481R, and, to a lesser extent R156E drastically reduce basal TIR1·IAA7, and auxin-driven TIR1·IAA7 and TIR1·IAA12 associations (Fig. 7a). Importantly, at high auxin concentrations the effect of the TIR1 mutations S201A, D481R and R156E on TIR1·IAA7 associations, weakens. We envision a scenario in which in a high auxin environment, an intact AUX/IAA degron is glued and engaged by TIR1, which overrides and probably compensates for the loss of transient or milder interaction interfaces.

To further determine whether the new TIR1·AUX/IAA interfaces are required for biological function in planta, we transformed *tir1-1* mutant plants with constructs expressing mutant versions of *TIR1* under the control of the TIR1 promoter. We introduced single and double mutations in TIR1 affecting putative engagement sites for the PB1 domain and KR motif of AUX/IAAs, including R156E, S199A, S201A, R220A, or D481R, and tested their ability to rescue the auxin-resistant phenotype of *tir1-1* plants in root-elongation assays (Fig. 7b). If the newly identified TIR1 sites facilitate transient interactions with AUX/IAAs, we reasoned the more informative effects would be those traceable at low auxin

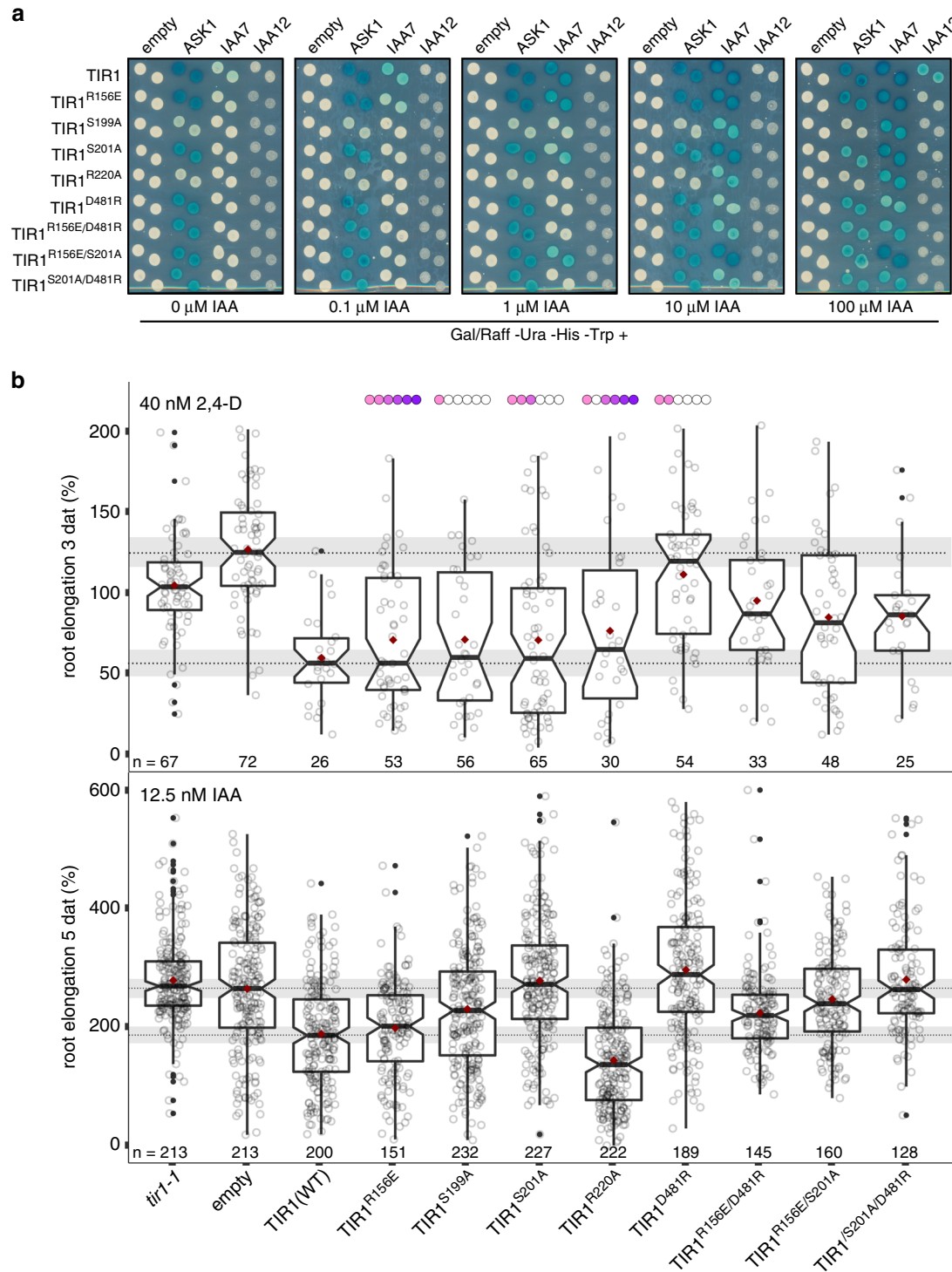

**Fig. 7 Mutations on novel putative AUX/IAA-binding sites in TIR1 impair auxin responses in vivo. a** Yeast two-hybrid (Y2H) interaction matrix for TIR1 wild type and TIR1-mutant versions carrying amino acid exchanges on relevant CAS-identified residues with ASK1, IAA7 and IAA12 at different auxin concentrations. **b** Box plots depicting root length of T1 *tir1-1* mutant plants carrying TIR1p:*tir1* mutant constructs. Transformed T1 seeds expressing RFP were selected by microscopy, and germinated in growth media. Five-day-old seedlings were transferred to growth medium containing either 40 nM 2,4-D or 12.5 nM IAA. Root elongation was traced on the day of transfer, and at days 3 and 5 after auxin treatment. *tir1-1* mutants are resistant to auxin treatment exhibiting long roots, and auxin sensitivity in *tir1-1* is restored by introducing a construct expressing wild type TIR1 under its own promoter. Numbers below the boxes correspond to the number of independent T1s (open circle's) analyzed per construct. Solid black horizontal lines represent median, dark red dots mean values and whiskers correspond to the upper and lower ~25% (1.5*IQR) of the data points. Outliers are shown as solid black dots. Shadow rectangles represent the notch size (~95% confidence interval) of the two reference data sets (empty vector and TIR1 wild type). Conservation of mutated residues in the TIR1/AFB1-5 F-box subclade is depicted as gradient colored circles above the plots (TIR1: light pink, AFB1-4: gradient of darker pink, AFB5: deep purple).

concentrations. Therefore, we transferred our transgenic lines to either a low concentration of natural IAA (12.5 nM), or a high concentration of synthetic auxin 2,4-D (40 nM). Compared to IAA, 2,4-D causes a sustained effect, as it accumulates over time in the cell[50]. As expected, a wild type version of TIR1 complements the auxin resistant *tir1-1* phenotype, while roots of *tir1-1* plants carrying the empty transformation vector, as a control, are blind to auxin, and continue elongating despite the treatment. Similarly, R156E and S199A restore wild type auxin sensitivity to seedlings treated with either 2,4-D or IAA for 3 to 5 days, respectively (Fig. 7b). This hints at those sites not having a prominent effect on TIR1 function in vivo. In contrast, S201A, D481R singles, and the double mutants R156E S201A and S201A D481R do not complement the root *tir1-1* phenotype of IAA treated plants (Fig. 7b). Although TIR1 S201 and R220 locate in the same cluster, they seem to affect TIR1 function differently. S201A complements the inhibitory effect of 2,4-D on root growth inhibition, indicating these plants might have been able to adapt to a sustained high auxin environment. R220A, on the other hand, confers dominant negative effects resulting in auxin hypersensitivity (Fig. 7b). In summary, we demonstrated the existence of two TIR1 amino acid clusters harboring S201, R220 and D481, essential for TIR1·AUX/IAA interaction interfaces, and TIR1 activity in vivo.

## Discussion

Auxin is perceived by TIR1/AFBs and their ubiquitylation targets the AUX/IAA transcriptional repressors. While TIR1 adopts a compact solenoid fold, AUX/IAAs appear flexible and modular in nature as they engage in various protein interaction networks[26,51]. A 13-aa degron motif in AUX/IAAs seals a ligand-binding groove in TIR1, and is secured by auxin in place. To date, we lacked information on whether additional physical interactions between TIR1 and AUX/IAAs influence conformation and fate. We also did not know whether these interactions facilitate the formation of the final auxin receptor complex by a two-dimensional search on the part of TIR1 on the AUX/IAA surface or vice versa. We found IAA7 and IAA12 exhibit a highly dynamic conformation on account of IDRs along their sequence, which seems to favor recruitment by TIR1. Computational and experimental studies have shown IDRs, such as those in AUX/IAAs, act as inter-domain linkers contributing to protein–protein interactions by exclusively or partially forming binding interfaces[17,52,53]. Capturing TIR1·IAA7 and TIR1·IAA12 ensembles by XL-MS allowed us to visualize AUX/IAAs IDRs embracing TIR1 and expanding their, known so far, interaction interfaces. Although IAA7 and IAA12 show differences on IDR content and length, both embraced TIR1 in a similar manner. While the AUX/IAA degron drives auxin-mediated TIR1·AUX/IAA interactions, we found evidence for the IDR upstream of the degron and the PB1 domain to engage in transient interactions with two specific clusters of amino acids at the C-terminal domain (CTD), and the N-terminal domain (NTD) of TIR1, respectively (Fig. 8). A directional embrace of TIR1 by an open-armed AUX/IAA, strengthened by degron-flanking IDRs, might additionally secure a TIR1·auxin·degron "click" (Fig. 8).

From the AUX/IAA standpoint, their local flexibility evidently shapes their conformation and accessibility when in complex with TIR1. Flexible IDRs in AUX/IAAs, as shown for IAA7 and IAA12, serve as variable calipers that measure the available distance between the KR motif and the core degron, and the degron and the PB1 domain, to properly and, with the right orientation, dock on TIR1. Our data also provided evidence for dynamic allosteric modulation of a TIR1·AUX/IAA complex by the folded PB1 domain and IDRs in AUX/IAAs. We could track positive but also negative cooperativity, due to the degron tail and PB1 domain combination, fine-tuning conformational states of TIR1·IAA7 and TIR1·IAA12 pairs, respectively. Further long-range, probable allosteric, effects are reflected into AUX/IAA turnover, when PB1 domain and degron tail act as one element (Supplementary Fig. 5).

The effects of cooperative allostery driven by IDRs in AUX/IAA proteins might not be limited to the TIR1·AUX/IAA inter-action, but rather influence the assembly into other complexes regulating auxin output signals[54]. It is therefore also possible that in response to fluctuating cellular auxin concentrations, transient TIR1·AUX/IAA interactions alter the energy landscape of AUX/IAA·TPL, AUX/IAA·ARF and AUX/IAA·AUX/IAA assemblies and/or possible decorations with PTMs. Future studies will tell whether IDRs in AUX/IAAs, and the recently described IDRs in ARFs, affect their protein assembly's localization or activity[55]. One can envision, IDR-driven cooperativity resulting in a multiplicity of allosterically regulated interactions within the auxin signaling pathway, where AUX/IAAs act as signaling hubs within the different complexes.

Within the *Arabidopsis* AUX/IAA protein family, nearly half of the degron tails are between 20 and 40 aa long and show high disorder probability (Supplementary Fig. 1). Seven of the 23 degron-containing AUX/IAAs (IAA19, IAA4, IAA6, IAA5, IAA1, IAA2, IAA15), however, carry a relatively ordered degron tail shorter than 20 amino acids (Supplementary Figs. 1 and 15). Is that specific length an evolutionary constraint for TIR1 association? Auxin-dependent gene regulation, and AUX/IAA proteins appear in the land plant lineage over 500 mya[28,56]. When comparing the proteins sequence of the ancestral AUX/IAAs in moss and *Marchantia*[57,58], we observed their degron tails are not much longer than the average degron tails (40 aa) of *Arabidopsis* AUX/IAAs, despite the overall length of these proteins being at least double that of angiosperm AUX/IAAs. It will be interesting to investigate whether degron tails length and disorder content are deeply conserved features for surface availability, and whether short degron tails (<20 aa) can still offer tailored positioning on TIR1. It remains also to be determined whether IDRs flanking the degron befit AUX/IAAs, particularly closely similar AUX/IAA ohnologs, with signatures that calibrate degron accessibility. Furthermore, the degron tail might generate an entropic force[59,60] that is fine-tuned, but also restricted, by IDR length, modulating binding of AUX/IAAs to TIR1. It remains to be established whether degron tails in different AUX/IAAs impact the interaction surface with TIR1, which we anticipate might translate into variability of binding kinetics.

Do structural features in TIR1 aid AUX/IAA positioning? Our data shows that is indeed the case. We found R220 located in cluster 1 to actively participate in TIR1·AUX/IAA associations in silico, in vitro, and in vivo. In fact, the TIR1 mutation R220A caused auxin hypersensitivity in *Arabidopsis* seedlings. Previously, D170E and M473L *tir1* mutant alleles showed faster AUX/IAA degradation, and increased transcription of auxin-responsive genes resulting also in auxin hypersensitivity[61]. Based on our biochemical and structural proteomics data, a few scenarios could explain the effect of R220A *TIR1* mutant allele. Thanks to its positive charge and size, R220 might play a sentry role for guiding the location of the disordered degron tail and the PB1 domain of AUX/IAAs on TIR1. Alanine-substituted R220 might result in a positional effect of the C-terminal portion of AUX/IAAs altering the exchange rates of different AUX/IAAs. Auxin-dependent, but also auxin-independent TIR1·AUX/IAA interactions could be expedited if the R220A conversion relaxes the positioning of the PB1, of at least a subset of AUX/IAAs. Most intriguingly, R220 is almost fully conserved among the members of the TIR1/AFB FBP subclade in *Arabidopsis* supporting its central role monitoring

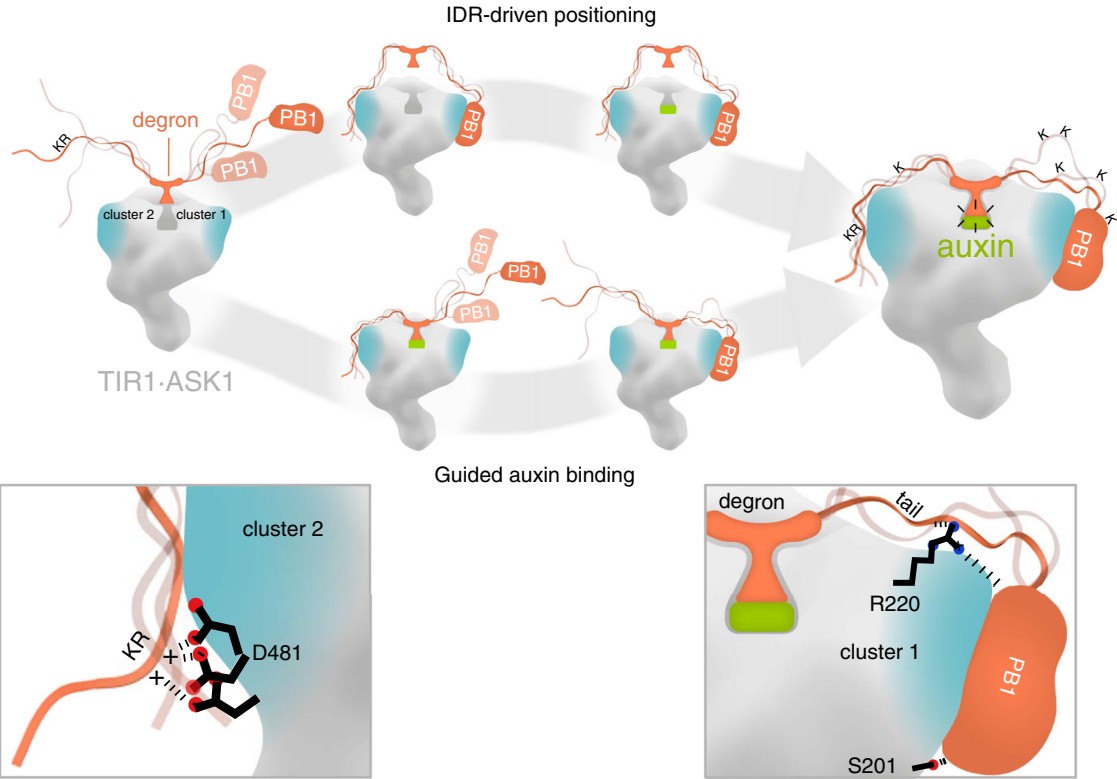

**Fig. 8 Model for ASK1·TIR1·AUX/IAA complex assembly fine-tuned by IDRs flanking the AUX/IAA degron.** The F-box protein TIR1 of the SCF[TIR1] E3 ubiquitin ligase recruits AUX/IAA targets for their ubiquitylation and degradation. The phytohormone auxin and a core degron in AUX/IAAs are essential for AUX/IAA recognition. Intrinsically disordered regions (IDRs) flanking the degron provide high flexibility and an extended fold to AUX/IAAs, influencing TIR1·AUX/IAA complex formation. At least two different routes are possible for dynamic AUX/IAA recruitment and UPS-mediated degradation: (i) auxin-triggered association between TIR1 and the AUX/IAA degron paves the way for positioning adjacent IDRs, which exposes ubiquitin acceptor sites for efficient ubiquitylation; (ii) transient auxin-independent interactions between IDRs, as well as the PB1 domain in AUX/IAAs, and two patches (clusters 1 and 2) of residues at opposite sides of TIR1, assist on auxin binding and offer tailored positioning. Residues R220 and S201 from cluster 1 (right zoom in) and D481 from cluster 2 (left zoom in) in TIR1 play a major role in TIR1·AUX/IAA complex formation. The residency time of an AUX/IAA target on TIR1, enables processivity of AUX/IAA ubiquitylation, and impinges on availability of IDRs as initiation sites for degradation by the 26S proteasome.

target recruitment (Fig. 7b). This data allowed us to postulate that the right positioning of the degron tail and the PB1 domain of AUX/IAAs on cluster 1 in TIR1 might have a favorable effect on auxin sensing, as part of the target recruitment mechanism (Fig. 8).

Particular stretches of amino acids with increased evolutionary conservation within disordered segments have been found to determine interaction specificity, acting as functional sites[62–64]. This seems to precisely apply to the region in AUX/IAAs upstream of the degron containing the auxin-responsive Lys-Arg (KR) dipeptide motif[35,65]. The KR exhibits a high level of conservation, and in addition to being part of a bipartite nuclear localization signal (NLS), the KR contributes to assembly of a TIR1·AUX/IAA auxin receptor complex and, probably as a result, is required for basal proteolysis in planta and AUX/IAA degradation dynamics[24,35,36,65]. Interestingly, the ability of the KR to act as auxin-responsive rate motif influencing AUX/IAA turnover, and the magnitude of this effect could only be correlated with the proximity of the KR to the degron[35,36]. How mechanistically could the KR exert an effect on TIR1 recognition and further AUX/IAA processing? Our findings lead us to propose an answer to a more than 10 year's long-standing question. As part of the AUX/IAA embrace to TIR1, we found the KR motif embedded in the IDR upstream of the degron confers alternative, and probably, first binding contacts with TIR1 (Fig. 8). We predict a high-IDR flexibility in the NTD of AUX/IAAs warrants a

necessary distance between the KR and the core degron for reaching distinct TIR1 contact sites, including D481 (Fig. 8). D481 is located in a negative charged patch in cluster 2 within the CTD of TIR1 (Fig. 8 and Supplementary Fig. 9). According to our XL data, the TIR1 exposed patch (incl., D481, S482, E459, or E506) comes into close proximity with the KR-containing IDR in AUX/IAAs making electrostatic interactions possible. We tested a reversed charge exchange for D481, and the resulting D481R abolished basal TIR1·IAA7 association, while weakening auxin-driven TIR1·IAA7 and TIR1·IAA12 interactions. Not only might a charge exchange lead to a repulsion of the AUX/IAA KR motif, but an Arg-replacement might displace and therefore slow down or prevent KR engagement. While TIR1 and AFB1 offer similar contact points to the KR in AUX/IAAs, AFB2, and AFB3 exhibit opposite charged residues (Lys) that however might still provide charge–charge interactions with a specific subset of AUX/IAAs. It remains to be determined whether this is an additional feature facilitating differential auxin sensing by distinct TIR1/AFBs·AUX/IAA co-receptor combinations[24].

The described interaction interfaces and structural disorder in AUX/IAAs appear also to be instrumental for processivity in ubiquitin transfer by the SCF[TIR1] E3 ubiquitin ligase. This is crucial as once an active E2-E3-target assembly has formed, spatial and geometric constraints such as distance and orientation relative to the E3-bound primary degron limit ubiquitylation surface and lysine selection for degradation[7]. AUX/IAA sequence

harbors a number of putative ubiquitin acceptor lysines (~9% total sequence) (Supplementary Fig. 15). Our data showed that not all of these sites are favorable for ubiquitylation. Downstream of the core degron, AUX/IAAs likely lend an attractive region for ubiquitin conjugation. We envision either the PB1 or the degron tail facilitating the accessibility of residues that undergo ubiquitylation. Upon TIR1·AUX/IAA interaction, IDRs either orient the PB1 domain-located lysines (e.g. IAA7) or act themselves as ubiquitylation acceptor sites as ubiquitin acceptor sites (e.g. IAA12). Properly positioned ubiquitin moieties at the suitable distance of an IDR, and an IDR with unbiased sequence composition as an initiation site will certainly impact efficient AUX/IAA degradation by the proteasome[66–68]. Hence, it will be imperative to shed light on where AUX/IAAs are ubiquitylated in vivo, and where exactly the proteasome initiates degradation relative to the ubiquitylation sites.

In summary, we unveiled an expanded network of TIR1·AUX/IAA interactions modulated by intrinsically disordered regions flanking the degron, and identified key residues for co-receptor formation and auxin perception. Our biochemical studies combined with a structural proteomics approach demonstrated IDRs in IAA7 and IAA12 harbor specific features that support TIR1·AUX/IAA interactions. In planta data confirmed these findings, and revealed a wider extent of TIR1·AUX/IAA interactions modulating auxin signaling, and likely enabling efficient ubiquitin transfer.

From a biological perspective, we evidenced that IDRs outside of a degron in ubiquitylation targets can participate, in particularly, basal interactions with an E3. We captured for the first time ensembles of a highly flexible ubiquitylation target and an SCF-type E3 ubiquitin ligase, identified novel interaction interfaces, and confirmed the relevance of specific interaction sites in vivo.

From a technical standpoint, XL-MS-based structural proteomics, which is yet to become widely regarded, offered a unique opportunity to visualize transient protein–protein interactions, otherwise difficult to capture. The gain in structural information, in combination with biochemical and in vivo validation opens up great opportunities to discover novel interaction interfaces and pinpoint new functional sites in a protein of interest. Additionally, our studies highlight the power of a combined experimental set-up for unraveling selection mechanisms in complex formation, and understanding how IDR-driven allostery might influence a complex signaling network.

## Methods

**Phylogenetic tree generation and secondary structure analysis**. Phylogenetic tree construction was done using Clustal Omega[69] with standard settings (Neighbor-joining tree without distance corrections), and the full length protein sequences of all *Arabidopsis* AUX/IAAs deposited at uniprot[https://www.uniprot.org/] (Supplementary Data 1). The constructed tree was visualized by iTOL[70] and manually edited. In silico disorder analysis was performed with the web-based IUPred2A tool[38] utilizing AUX/IAA protein sequences. The resulting disorder probability was used to categorize each residue as either ordered (<0.4), intermediate (0.4–0.6), or disordered (>0.6). Same analysis was carried out for all AUX/IAA proteins excluding the PB1 domain (for reference, the conserved VKV motif was earmarked as the start of the PB1 domain). Residues of each category were plotted using R. IAA7 and IAA12 disorder predictions were additionally carried out using SPOT[71] and PrDOS[72] algorithms with standard settings. Hydropathy plots were generated via Expasy-linked ProtScale[73,74] using the Kyte-Doolittle method[75].

**Protein purification**. ASK1·TIR1 complex was purified from Sf9 cells as described earlier[23] with minor changes. In brief, ASK1 was co-purified with GST-TIR1 using GSH affinity chromatography (gravity flow) and anion chromatography (MonoQ) followed by tag-removal and a final size-exclusion chromatography (SEC) step (Superdex 200) using an ÄKTA FPLC system.

AUX/IAA proteins, including chimeric versions, were expressed as GST-tagged proteins in *E.coli* and purified using GSH affinity chromatography, including a high-salt wash (1 M NaCl) and gravity flow anion exchange chromatography (Sepharose Q). For circular dichroism, the GST-tag was removed on the GSH

column matrix with thrombin, and fractions containing AUX/IAAs were briefly concentrated, passed over a benzamidine column, and further purified using a Sephacryl S-100 column (SEC) with an ÄKTA FPLC system. This step was carried out using the CD measurement buffer (see CD measurement section) for buffer exchange.

**Size exclusion chromatography and size calculations**. The last protein purification step was used to simultaneously determine the Stokes radii of AUX/IAAs in CD buffer (10 mM KPi pH 7.8; 150 mM KF; 0.2 mM TCEP). The HiPrep 16/60 Sephacryl S-100 high-resolution column was calibrated using gel filtration standards (Bio-Rad, Cat. #151-1901) with added bovine serum albumin (BSA) before the runs. Stokes radii for the globular known reference proteins were calculated as described[76]. The Stokes radii of AUX/IAA variants were calculated from the resulting calibration curve equation based on their retention volume ($n = 4$–$10$).

**Circular dichroism (CD) measurements**. After purification, including tag-removal and size-exclusion chromatography, AUX/IAAs were concentrated and adjusted to 2.5–5 µM in CD buffer. CD measurements were carried out on a Jasco CD J-815 spectrometer and spectra were recorded from 260 nm to 185 nm as 32 accumulations using a 0.1 nm interval and 100 nm/min scanning speed. Cell length was 1 mm and temperature was set to 25 °C. All spectra were buffer corrected using CD buffer as a control and converted to mean residual ellipticity (MRE). Reference spectra for a disordered (MEG-14; PCDDBID: CD0004055000 [https://pcddb.cryst.bbk.ac.uk/deposit/CD0004055000/?files=&dl]), a beta-sheet (BtuB; PCDDBID: CD0000102000 [https://pcddb.cryst.bbk.ac.uk/deposit/CD0000102000]) and an alpha-helical protein (amtB; PCDDBID: CD0000099000 [https://pcddb.cryst.bbk.ac.uk/deposit/CD0000099000]) were used.

**[3H]-labeled auxin binding assay**. Radioligand binding assays for determining dissociation constants of auxin receptors[77] were performed using purified ASK1·TIR1 protein complexes, GST-tagged AUX/IAAs (incl. chimeric AUX/IAAs) and [3H]IAA with a specific activity of 25 Ci/mmol (Hartmann Analytic). Final protein concentrations in a 100 µL reaction were 0.01 µM ASK1·TIR1 complex and 0.3 µM AUX/IAAs. Complexes were allowed to form 1 h on ice, shaking. For non-specific binding controls, reactions contained additionally 2 mM cold IAA. Data was evaluated with GraphPad Prism v 5.04, and fitted using the "one site total and non-specific binding" preset.

**LexA yeast two-hybrid assays**. LexA-based yeast two-hybrid assays were performed using mated yeast strains EGY48 + pSH18-34 and YM4271 transformed with either LexA DBD-fusions of TIR1 or tir1 mutants in the pGILDA vector; or AD-fusions of ASK1, IAA7, IAA12, or iaa7/12 chimeras in the pB42AD vector (GoldenGate system, Supplementary Data 1). For each assay, same count of yeast cells ($OD_{600} = 0.4$ or $0.8$ for IAA12(-like)) were spotted on selection media (Gal/Raff–Ura –His–Trp) containing BU salts (final: 7 g/L $Na_2HPO_4$, 3 g/L $NaH_2PO_4$, pH 7), X-Gal (final 80 mg/L) and the given auxin (IAA) concentration. Plates were incubated at 30 °C for several days and constantly monitored. Expression of chimeric AUX/IAAs and TIR1 mutants in yeast was confirmed via immunoblot analysis on lysates from haploid yeast. Fifty milliliters liquid selection medium (Gal/Raff -Ura -His or -Trp) were inoculated with an 1/25 volume overnight culture and grown to $OD_{600} \approx 0.6$. Cells were harvested, washed with distilled water and lysed in 200 µL lysis buffer (0.1 M NaOH, 2% β-mercaptoethanol, 2% sodium dodecyl sulfate, 0.05 M EDTA, 200 µM PMSF, 1 mM benzamidine, Roche protease inhibitor cocktail) at 90 °C for 10 min. After neutralization with 5 µL 4 M sodium acetate for 10 min at 90 °C, 50 µL 4X Laemmli was added and samples were separated via SDS-PAGE and immunoblotted (anti-HA(F-7): Santa Cruz Biotechnology (sc-7392; 1:1000), anti LexA: abcam (ab14553; 1:500), anti-Tubulin (YL1/2): abcam (ab6160; 1:5000), anti-rabbit-AP: Sigma-Aldrich (A3687; 1:10000), anti-mouse-AP: Sigma-Aldrich (A2179; 1:10000)).

**In vitro reconstitution of Ub-conjugation (IVU)**. In vitro ubiquitylation (IVU) reactions[34] were prepared as follows: Two protein mixtures (mix A and mix B) were prepared in parallel. Mix A contained 50 µM ubiquitin (Ub; fluorescein-labeled $Ub^{S20C}$: $Ub^{K0}$; 4:1 mix), 0.2 µM 6xHis-UBA1 (E1) and 2 µM 6xHis-AtUBC8 (E2) in reaction buffer (30 mM Tris-HCl, pH 8.0, 100 mM NaCl, 2 mM DTT, 5 mM $MgCl_2$, 1 µM $ZnCl_2$, 2 mM ATP). Mix B contained 1 µM Cul1·RBX1, 1 µM ASK1·TIR1, and 5 µM AUX/IAA protein in reaction buffer. Mix B was aliquoted and supplemented with IAA to reach the indicated final concentration. Mixtures A and B were separately incubated for 5 or 10 min at 25 °C, respectively. Equal volumes of mix A and B were combined, aliquots were taken at specified time points, and reactions were stopped by denaturation in Laemmli buffer. IVUs with chimeric AUX/IAAs were carried out 1 h with 1 µM IAA. Immunodetection of Ub-conjugated proteins was performed using polyclonal anti-GST in rabbit (1:20,000; Sigma, G7781) antibodies combined with secondary anti-rabbit Alexa Fluor® Plus 647 antibody (1:20,000; Thermo Fischer Scientific, A32733). Detection was performed with a Typhoon FLA 9500 system (473 nm excitation wavelength and LPB filter for fluorescein-labeled ubiquitin signal detection and 635 nm excitation wavelength and LPR filter for GST signal).

Quantification of ubiquitylated AUX/IAAs was achieved by using ImageQuant TL software automatic lane detection of in-gel fluorescein signals above unmodified GST-IAA7 and GST-IAA12 proteins (~50 kDa). As the signal for ubiquitylated AUX/IAAs increase, the signal for unmodified GST-IAA7 and GST-IAA12 fusion proteins decreases. This was quantified after blotting and immunodetection using the Alexa Fluor 647 signal, and automatic band detection. All signals were background subtracted (rubberband method).

**LC-MS analyses of IVU reactions.** Three sets of IVUs, corresponding to three biological replicates, were performed on consecutive weeks using AUX/IAA proteins from different batch preparations. After 30 min, IVUs were stopped by denaturing with urea, reduced with DTT and alkylated with iodoacetamide. Trypsin digestion was carried out overnight at 37 °C. Upon quenching and desalting, peptides were separated using liquid chromatography C18 reverse phase chemistry and later electrosprayed on-line into a QExactive Plus mass spectrometer (Thermo Fisher Scientific). A Top20 DDA scan strategy with HCD fragmentation was used for MS/MS peptide sequencing. Ubiquitylated residues on identified peptides were mapped using GG and LRGG signatures (as tolerated variable modifications) from using both the Mascot software v2.5.0 (Matrix Science) linked to Proteome Discoverer v1.4 (Thermo Fisher Scientific), and the MaxQuant software v1.5.0.0.

**Crosslinking (XL) reactions and LC-MS analyses.** DSBU (ThermoFisher) XL reactions containing either 4–5 μM of ASK1·TIR1, and 5 μM IAA7[BM3] or IAA12[BM3] or 10 μM IAA7[BM3] or IAA12[BM3] alone were incubated for 1 h at 25 °C. Proteins were pre-incubated 15 min in the presence or absence of 10 μM auxin (IAA) before addition of 1 mM DSBU (100 molar excess). After TRIS quenching, samples were sonicated in the presence of sodium deoxycholate, reduced with DTT, and alkylated with iodoacetamide. Alkylation was further quenched by DTT, samples were incubated with trypsin overnight at 37 °C, and protein digestion was stopped with 10% TFA.

Upon centrifugation (5 min 14,000 x $g$), proteolytic peptide mixtures were analyzed by LC/MS/MS on an UltiMate 3000 RSLC nano-HPLC system coupled to an Orbitrap Q-Exactive Plus mass spectrometer (Thermo Fisher Scientific). Peptides were separated on reversed phase C18 columns (trapping column: Acclaim PepMap 100, 300 μm × 5 mm, 5 μm, 100 Å (Thermo Fisher Scientific); separation column: self-packed Picofrit nanospray C18 column, 75 μM × 250 mm, 1.9 μm, 80 Å, tip ID 10 μm (New Objective) or μPAC™ 200 cm C18 (Pharmafluidics). After desalting the samples on the trapping column, peptides were eluted and separated using a linear gradient from 3% to 40% B (solvent A: 0.1% (v/v) formic acid in water, solvent B: 0.08% (v/v) formic acid in acetonitrile) with a constant flow rate of 300 nL/min over 90 min. Data were acquired in data-dependent MS/MS mode with stepped higher-energy collision-induced dissociation (HCD) and normalized collision energies of 27%, 30%, and 33%. Each high-resolution full scan ($m/z$ 375 to 1799, $R = 140,000$ at $m/z$ 200) in the orbitrap was followed by high-resolution product ion scans ($R = 17,500$) of the ten most intense signals in the full-scan mass spectrum (isolation window 2 Th); the target value of the automated gain control was set to 3,000,000 (MS) and 200,000 (MS/MS), maximum accumulation times were set to 100 ms (MS) and 250 ms (MS/MS) and the maximum cycle time was 5 s. Precursor ions with charge states <3+ and >8+ or were excluded from fragmentation. Dynamic exclusion was enabled (duration 60 s, window 3 ppm).

**Data analysis of crosslinked (XL)-peptides.** For XL analysis, mass spectrometric *.raw files were converted to mzML using Proteome Discoverer 2.0. MeroX analysis was performed with the following settings: Proteolytic cleavage: C-terminal at Lys (blocked as XL site) and Arg with max. 3 missed cleavages, peptides' length: 5 to 30, static modification: alkylation of Cys by IAA, variable modification: oxidation of M, crosslinker: DSBU with specificity towards Lys, Ser, Thr, Tyr, and N-termini, analysis mode: RISE-UP mode, precursor mass accuracy: 5 ppm, product ion mass accuracy: 10 pm (performing mass recalibration, average of deviations), signal-to-noise ratio: 1.5, precursor mass correction activated, prescore cutoff at 55% intensity, FDR cutoff: 5%, and minimum score cutoff: 70. All analyses included the cRAP database sequences. Decoy database was generated using shuffled sequences with kept protease sites. Shown in Fig. 4 are all detected intramolecular XL and all ASK1·TIR1 XLs. Sequences of IAA7 and IAA12 contain 5 or 2 additional amino acids at the N-terminus, respectively. Detailed results can be found in Supplementary Data 3 and 4. For further analysis only inter-protein XLs between TIR1 and AUX/IAAs found in at least 2/3 (IAA7) or 3/4 (IAA12) experiments were considered.

**XL-based docking using HADDOCK and DisVis analysis.** Comparative models of IAA7 and IAA12 PB1 domains were created using multi-sequence-structure-alignments (PIR formatted) as input for MODELLER 0.921[78]. Input files, alignment files and derived models are provided in the Supplementary Data 5 (modeller_files.tgz). In addition, the C-terminal helix of both IAA7 and IAA12 PB1 domains were modeled de novo and subsequently added to the structure (resulting pdb: C-ter). The generated models (ten C-terminal helix variants) were incorporated for the HADDOCK-based docking together with the available

ASK1·TIR1 structure (PDB code https://www.rcsb.org/structure/code: 2P1Q [http://www.rcsb.org/structure/2P1Q], resolution: $R = 1.91$ Å)[23]. HADDOCK parameter files are provided in the Supplementary Data 6 (haddock_files).tgz. A detailed description on how to prepare pdb files and incorporate distance restraints can be found elsewhere[79]. Formatted pdb files were uploaded to the HADDOCK server[80,81] using guru access level. To incorporate restraints, we used known distances reported for DSBU and albumin XLs from our data sets (Supplementary Table 1 and Supplementary Data 3 and 4). Accordingly, we further added a distance restraint (degron tail restraint) corresponding to the degron tail length calculated as described in ref. [82]. Here, the theoretical Stokes radii of a given peptide for different folding states (min: folded; max: disordered) are calculated and used as restraints. For each complex docked, 10,000 rigid body docking structures were generated followed by a second iteration (400 best structures). Finally, 200 models/structures were water refined (explicit solvent) and clustered (FCC[83] at 0.6 RMSD cutoff).

Using the same restraints, the possible conformational docking space of the PB1 domains was searched, and visualized using DisVis[49,84,85] with standard parameters (Supplementary Data 7 (DisVis_only_files).tgz). In addition, in order to validate the derived models, we performed a docking with HADDOCK including both, the distance restraints shown in Supplementary Table 1, and active residues calculated by DisVis. In brief, the restraints were used to generate the possible conformational docking space of the PB1 domains, followed by calculation of active residues, based on their interaction propensities using DisVis. DisVis considers those residues as active most contacted in the solutions consistent with the provided distance restraints. Those residues with an interaction propensity higher than 1.0 were selected, and subsequently used as active residues for the docking with HADDOCK under the general definition of ambiguous interaction restraints[86]. Parameter files used for this docking and final structures are provided as Supplementary Data 8 (disvis_haddock).tgz. The combination of distance restraints and DisVis-calculated active residues showed high restraint violation energies and the results from this approach were not further used. PyMOL™ (Version 2.1) and UCSF Chimera[87] were implemented for image creation.

**Molecular dynamic simulations of protein–protein complexes.** One refined structure of each group, derived from the XL-based docking by HADDOCK incorporating the disorder restraint (two groups for TIR1·IAA7[PB1]; three groups for TIR1·IAA12[PB1]), was used as starting structure for MD simulations (cluster1_1 and cluster2_1 from haddock_files.tgz/haddock_files/IAA07/With_disorder_restraint/; cluster1_1, cluster2_1 and cluster3_1 from haddock_files.tgz/haddock_files/ IAA12/ With_disorder_restraint/, both from Supplementary Data 6 (haddock_files).tgz). The five structures were prepared using structure preparation and protonate 3D (pH = 7.5) modules and subsequently minimized with AMBER10 force-field[88] in MOE 2019.0101 (Chemical Computing Group Inc., Montreal, Quebec, Canada).

Molecular dynamic simulations were performed with the GROMACS software package (version 4.6.5)[89]. The parameters corresponding to the proteins were generated with AMBER99SB-ILDN force-field[90] and TIP3P explicit solvation model[91]. Electro-neutrality was guaranteed by adding Na$^+$ and Cl$^-$ ions into the unit cells at an appropriate ratio to reach a final NaCl concentration of 0.2 mol/L. The protocol employed here to perform MD simulations involves prior energy minimization (EM) and position-restrained equilibration, as outlined by Lindahl[92] for lysozyme in water. Newton's equation of motion for the position-restrained equilibration was solved using the leap-frog integrator[93], with a time step of $\Delta t = 2$ fs for a total time of 50 ps (25,000 integration steps). The system was simulated at constant temperature and pressure of 310 K and 1 atm, respectively. The Berendsen algorithm[94] for the pressure and Velocity rescaling[95] for the temperature with time constant ($\tau$) of 3 ps and 0.1 ps, was respectively implemented[96]. Obeying the Maxwell–Boltzmann distribution from 50 to 310 K[96] random initial velocities were assigned to each atom prior to the MD simulations.

Once the system was equilibrated, we proceeded to the productive dynamic simulation without position restraint[97] for 20 ns. The system simulation was carried out at $T = 310$ K and $p = 1$ atm. The Parrinello-Rahman coupling algorithm[98,99] was used to keep pressure constant with a time constant ($\tau$) of 1 ps[96]. The temperature, non-bonded interaction and time step were controlled or set up similarly as in the equilibration run. The snapshots of all runs were saved each 10 ps. Molecular dynamics parameter files (mdp), and the minimized and equilibrated starting structure for each run, are provided in the Supplementary Data 9 (MD).tgz. Detailed MD simulation method is included in Supplementary Methods.

**Effective binding free-energy calculations using MM-GBSA.** The effective binding free-energy ($\Delta G_{eff}$) of the protein–protein complexes formation was calculated using MMPBSA.py from Amber18 package employing the MM-GBSA method[100]. We followed the single trajectory approach, in which the trajectories for the free proteins were extracted from that of the protein–protein complexes. $GB^{OBC1}$ and $GB^{OBC2}$ implicit solvation models were employed[100]. The $\Delta G_{eff}$ values were obtained every 10 ps from the productive MD simulation (20,000 ps). We calculated the cumulative mean (also referred to as accumulated mean) for each of the 2000 $\Delta G_{eff}$ values. We computed the accumulated mean for each position by summing over all previous values and dividing by their number.

Energetically relevant residues (hot-spots) at the interfaces of TIR1·AUX/IAA[PB1] complexes were predicted by using the per-residue effective free-energy decomposition (prEFED) protocol implemented in MMPBSA.py[100]. Hot-spot residues were defined as those with a side-chain energy contribution ($\Delta G_{SC}$) of $\leq -1.0$ kcal/mol. We used computational alanine scanning (CAS)[100] to further assess per-residue free-energy contributions. Alanine single-point mutations were generated on previously identified hot-spots from the prEFED protocol. Both prEFED and CAS protocols were performed from the last 10 ns of the MD simulation.

**Plant materials and root-elongation assays**. Transgenic *Arabidopsis thaliana* plants expressing mutated TIR1 versions (tir1cds[(mut)]) driven by the TIR1 promoter (TIR1p) were generated using Gateway cloning. A ~2.3 kB TIR1 promoter fragment was amplified and subcloned in a pUC57-based entry vector using *Xma*I and *Kpn*I sites. Similarly, TIR1 cDNA fragments, either wild type or carrying mutations in KR- or PB1-binding sites were amplified from the pGILDA-TIR1 or pGILDA- tir1cds[(mut)] yeast expression constructs[24]. A list of primers utilized in this study has been provided as Supplementary Data 1. Primers added Gateway *attB* recombination sites to clone inserts into pDONR221[TM]/ZEO (Invitrogen). Final constructs were obtained using a double Gateway reaction of the respective Entry clones and pEN-4 entry vector containing the TIR1 promoter, into the destination vector pDEST (pEDO 097 (4 ccdb-2)) TIR1p:tir1cds[(mut)]. Nine different TIR1p:tir1cds[(mut)] constructs and TIR1p:TIR1cds[(wt)], as a complementation control, were introduced by Agrobacterium-mediated transformation into *tir1-1* mutant plants. Transformed seeds (T1s) expressing red fluorescence protein (RFP) were selected by fluorescence microscopy, surface sterilized, and directly sowed in ½ MS medium with 1% sucrose. After seed stratification for 3 days at 4 °C, seedlings were grown at 22 °C under long day (LD) conditions (16 h light, 8 h dark) and 90 µE/m²/s of light for 4 days. For auxin treatments, about 250 T1 seedlings per construct were transferred to vertical plates containing ½ MS growth medium (with 1% sucrose) supplemented with auxins, either 12.5 nM IAA (indole 3-acetic acid) or 40 nM 2,4-D synthetic analog (2,4-dichlorophenoxyacetic acid). Seedlings were grown at 22 °C under LD conditions, and root elongation was traced up to 5 days after transfer to auxin plates.

**Reporting summary**. Further information on research design is available in the Nature Research Reporting Summary linked to this article.

## Data availability

All data generated in this study has been made available either in the Source Data File, via the respective repository entry, or is provided as separate files. Mass spectrometry proteomics data have been deposited to the ProteomeXchange Consortium via the PRIDE partner repository with the data sets identifiers: PXD015285 [https://www.ebi.ac.uk/pride/archive/projects/PXD015285] (XL-MS) and PXD015392 [https://www.ebi.ac.uk/pride/archive/projects/PXD015392] (ubiquitylation site identification data). All other information supporting the findings of this study is available from the corresponding author upon reasonable request.

## Code availability

All code used in this manuscript is published and described in the corresponding parts of the manuscript.

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

## Acknowledgements

We thank Wolfgang Brandt for initial in silico models of AUX/IAA PB1 domains, and Silvestre Marillonet for the design of constructs for Golden Gate Technology. Thanks to Steffen Abel, Elisabeth Chapman, and Claus Schwechheimer for providing input to the manuscript. This work was supported by the Deutsche Forschungsgemeinschaft (DFG: research project CA716/2-1, and Research Training Group RTG2467), and core funding of the Leibniz Institute of Plant Biochemistry (IPB).

## Author contributions

M.N., E.M.C., and L.I.A.C.V. prepared the manuscript and designed experiments. M.N. performed biochemical experiments and analyzed the data. M.N., C.I., and C.H.I. carried out XL-MS experiments and data analysis. P.K. and M.N carried out all HADDOCK-based

approaches including DisVis, and E.M.C. computational calculation and simulations. M.N., A.H., and V.W. generated Y2H constructs and performed the assays. M.N. designed and executed ubiquitylation experiments, and together with W.H. analyzed mass spectral data of ubiquitylation sites. S.S. and M.Z. carried out ratiometric experiments and analyzed the data. M.N., E.M.C., V.W., and L.I.A.C.V. generated and analyzed *Arabidopsis* transgenic lines. E.M.C., C.I, C.I., P.K., and A.S. provided input to the manuscript. All authors approved the intellectual content.

## Competing interests
The authors declare no competing interests.
