## [Peer review file · Nature Communications]

Reviewers' comments:

Reviewer #1 (Remarks to the Author):

This manuscript combined an impressive array of methods to characterize the interaction between two AUX/IAA proteins with TIR1 in the context of auxin sensing. The authors identified new binding interfaces between AUX/IAA and TIR1 that were not present in the crystal structures in the 2007 Nature paper. Crosslinking MS experiments and mutagenesis data nicely corroborate to support the importance of the new binding sites, which allowed the authors to propose a structural model of the complex. The biochemical data are very solid and the findings are highly novel. These findings provided structural basis for understanding the differential auxin binding of the co-receptor complexes with different AUX/IAA components. However, a major caveat of the manuscript is that the paper solely relied on yeast two hybrid binding assays and in vitro data. Therefore, the manuscript would be much strengthened if the authors can show that the identified new interfaces actually affect AUX/IAA degradation in response to auxin in cells. For example, the mutations of TIR1 (D170K and S172A) t dramatically abolished binding of AUX/IAA in Y2H experiment, do they affect AUX/IAA degradation at all ? Same questions could be asked for the AUX/IAA chimera or deletion mutants. Adding the functional data would make the paper more relevant to plant genetic studies.

More specific comments are below:

Fig 1C, disorder/beta sheet/alpha helical labels are difficult to distinguish, using a solid line ?

Fig. 2b-2d, the authors need more repeats to generate statistics of the Kd values to draw conclusion about whether the different mutants indeed bind auxin with different affinity. This kind of measurement usually has a high variations. Also again, the curves are difficult to distinguish, please use different colors.

Supplementary Fig 1: would be nice to label the position of the degron in each panel.

Fig. 1C, since IAA7 and IAA17 are mostly disordered, one would expect that it would be hard to purify the proteins. A coomassie staining gel of the purified proteins would be required to show the purity/integrity of the proteins. In addition, since the proteins are purified from E. coli, one cannot rule out that the prokaryote system might not fold the protein properly. Have the authors tried AUX/IAA expression in insect cells as TIR1-ASK1 ?

Page 11 line 191, there is an extra "12" that needs to be removed.

Page 12 line 205, Should be "Supplementary Fig. 4d".

Page 12 line 215 Should be "Taken together"

Supplementary Fig 8b, are these quantifications from several independent experiments, could the authors put p-values on the figure to indicate statistical significance. This is even more relevant given that in Figure 3, it really difficult to see the difference of the fluorescence intensity just by eyes, for example between 77777 and 777127. Also the orders of the samples between Supplementary Fig 8 and Figure 3 are different. Please make sure about the labeling and stick to the same order.

Fig. 5a-b, What are the restrains indicated in the x-axis ? Why is the additional degron tail restraint only restricted the accessible conformations in the last column group ?

Reviewer #2 (Remarks to the Author):

Auxin is the most critical hormone in plant biology, driving all cell division and cell elongation events. Despite its great importance, its signal transduction pathway is seemingly simple, with a repression – depression paradigm, driven by auxin-mediated interactions between the SCFTIR1 complex and an Aux/IAA repressor, ruling auxin responses.

In this manuscript, the authors use a variety of approaches to resolve the topology of TIR1 contact points with IAA7 and IAA12, with a focus the C-terminal intrinsically disordered regions of these proteins.

Overall, this work will provide important details to the auxin community that may also be relevant to interactions among other F-boxes with their targets.

I have a few minor comments for improving this manuscript.

1- Line 264– It seems strange to me that the authors did not cite Gilkerson (2015) who showed by mutational analysis that Aux/IAA polyubiquitylation is flexible and can take place on many potential residues. At the very least, this should be discussed in the context of their work.

2- Line 346 – The authors make a strong statement about the numbers of assemblies detected for IAA12bm3 and IAA7bm3 and distribution of inter-protein crosslinks. Based on difficulties in identifying all peptides in mass spec, I am not comfortable with such a strong conclusion – the authors should soften their statements on this.

3- The theme of hugging and kissing and embracing throughout the discussion to describe TIR-Aux/IAA interactions seems inappropriate. I am guessing that a “hug” and a “kiss” are meant to depict different types of interactions, but it is difficult to tell. I would suggest using more specific, non-colloquial, language.

Reviewer #3 (Remarks to the Author):

Authors used interdisciplinary approach and combined an in silico analysis, biochemical analysis of recombinantly expressed proteins, the yeast two hybrid system (Y2H) for qualitative assessment of auxin co-receptor assembly as well as ubiquitylation dynamics of selected proteins (IAA7 and IAA12). Using spectroscopic (CD) and diffusional experiments in combination with recombinantly expressed proteins they found that N terminal part of these proteins is disordered with an extended structure in solution.

Using Y2H approach of chimeric proteins they investigated impact of intrinsic disordered segments as well as of the compact PB1 domain of AUX/IAA (IAA7 and IAA12) proteins on their interaction with TIR1, auxin independent, as well as auxin-triggered.

They stated that they confirmed the postulated interdependency of the degron tail and PB1 domain, but I am not convinced since chimeric proteins without PB1 nicely bind to TIR1, also, there is no experimental data (or I overlooked them?) on the PB1 - TIR1 binding. They point to additive and separate effects of each, disordered degron tail and the PB1 domain, on auxin independent and auxin-triggered TIR1 interaction. Furthermore, they postulated that AUX/IAA ubiquitylation favorably occurs at the solvent exposed regions in IAA7 and IAA12, when they are recruited by TIR1. Many assumption but without firm evidences.

Based on the structural proteomics approach authors assumed AUX/IAs IAA7 and IAA12 exhibit flexible conformations in solution (intra-protein cross-links), and adopt an extended fold when

bound to TIR1. But the cross-linking reactions were performed for chimeric (PB1 missing domain) proteins, and to get more convincing assumption the data for the native proteins are needed.

Y2H

I am not an expert, but from the data shown, it is hard to follow the statements given in the manuscript. For example:

1. The statement on IAA7 and IAA12 – TIR1 binding affinity ' Expression of the β -galactosidase reporter indicates stronger interaction of TIR1•IAA7 than TIR1•IAA12.' Is not correlated with Fig. 2a, since according to the data shown it seems that TIR1•IAA12 bind only at VERY high IAA conc. ($> 100 \mu$ M).
2. Author wrote that they docked homology-modeled PB1 domains of Arabidopsis IAA7 and IAA12 to the ASK1•TIR1 complex, applying distance restraints based on the cross-linking data. The mentioned distance restraints are related to the regions out of PB1, and it is not clear how authors modeled 3D structure of these regions?

Ubiquitylation

3. The statement: 'IAA12~ubiquitin conjugates were much less abundant than IAA7 after 30 min incubation' can not be derived from the data represented in Fig3a. From Fig. 3a it seems that they (IAA7 and IAA12) are similarly ubiquitinated.

Cross-linking

How long is the cross linker used? Are there data on its influence on the protein conformation?

Figure 6:

4. Figs 6a and 6b - From the ΔG graphs it seems that the affinity of PB1(IAA7) for TIR1 is lower than that of PB1(IAA12) – TIR1, while according to the experiments IAA7 binds to TIR1 with higher affinity than IAA12
5. The following statement: 'Mutations R156E, as well as S201A, and S205A either abolished or drastically impaired basal TIR1•IAA7 and auxin-driven TIR1•IAA7 and TIR1•IAA12 associations, without affecting ASK1•TIR1 assembly.' could not be deduced from Fig 6 e. According to the figure the statement is only partly true for R156E mutants of IAA7 at low auxin conc. (0.1 and 1 μ M), but for IAA12, as well as for IAA7 at higher IAA conc, there is no significant difference with respect to the WT proteins.6.

Fig2b, the affinity curve for IAA/bm3 have not reached the saturation (plateau)

The computational methods used in this manuscript is really strange:

1. Authors performed homology modelling of IAA7 and IAA12 PB1 domain, but what is with the other regions of the protein (DI, linker, degron tail?). How many and what structures were used as templates? What was the quality of the obtained models, how big is the 3D structure?
2. Authors wrote that 'how to incorporate distance restraint' as input for HADDOCK can be found elsewhere (they did not give a reference), but these restraints are related to regions out of PB1 while they have homology model only of PB1 domain. How they combined distance restraint data with non existing 3D structure for the purpose of docking?
3. Molecular modelling procedure: Structures (PB1(IAA7)-TIR and PB18IAA12)-TIR were optimized in MOE (using AMBER10 ff),
But there is no reference. As far as I know AMBER10 ff does not exist! Further on, the simulations were performed in GROMACS with AMBER99SB-ILDN force-field (the energy minimization and MD simulations should be performed with the same ff). Authors states that 'electro-neutrality was guaranteed with a NaCl concentration of 0.2 mol/L'. What was net charge of the system! Electro-

neutrality can not be achieved by adding the same amount of Na⁺ and Cl⁻, but, by adding certain number of either Na⁺, or Cl⁻ ions, depending on the charge of the system.

Nothing about time step, ensemble, barostat, thermostat ...

Finally, the binding free energies were calculated by MMPBSA.py from Amber18 package. Why, not in Gromacs, since it was used for MD simulations, and it has his own MMPBSA tool. Why usage of so many different program packages (and ff), when everything could be done within either GROMACS the AMBER18 package?

Figures 6 a & b: Authors wrote: 'The accumulated mean value of ΔG_{eff} were obtained every 10 ps' What does mean 'accumulated'? It should be the 'mean value of ΔG_{eff} ' (or running average). What was the frequency of sampling?

In general the sentences are too long, awkward and imprecise, for example:

'The residency time of an AUX/IAA target on TIR1, when assembled in an SCF-complex, propels processivity of AUX/IAA ubiquitylation...'

(How time can propels something, maybe correlate OR define ...)

'IDRs flanking the degron provide high flexibility and an extended fold to AUX/IAAs'

Should be 'ensure' (or enable..)

' ΔG_{eff} values over 20 ns identifying and refining stable TIR1•AUX/IAA PB1 complexes'...

Identifying is OK, but the calculated ΔG_{eff} values do not refine structures!

Also, some citations are (in my opinion) not adequate, for example for the statement 'Under low auxin concentrations, AUX/IAAs repress type A AUXIN 53RESPONSE FACTORS (ARF) transcription factors via physical heterotypic interactions through their type I/II Phox/Bem1p (PB1) domain'

authors cite: Liscum, E. & Reed, J. Plant Mol Biol (2002) 49: 387,

instead of

Tiwari S.B., Hagen G., Guilfoyle T.J. (2004). Plant Cell 16: 533-543

OR

Tom J. Guilfoyle Plant Cell. 2015 Jan; 27(1): 33-43.

Reviewer #4 (Remarks to the Author):

The paper by Niemeyer et al presents an interesting hypothesis on the role of intrinsically disordered regions in the auxin-mediated interaction between INDOLE-3-ACETIC ACID proteins (transcriptional repressors) and F-BOX PROTEINS TRANSPORT 57 INHIBITOR RESPONSE 1 (TIR1)/AUXIN SIGNALING F-BOX 1-5 (AFB1-5). The authors claim as their main conclusion that the intrinsic flexibility of AUX/IAAs ultimately affects the ubiquitination system, and in turn proteasome recruitment. With this angle, we believe the manuscript has considerable impact due to the crucial role of the ubiquitin-proteasome system which, in combination with the effect of Auxin concentration, is crucial to regulating plant growth.

The work represents additionally a serious effort in its experimental setup, in which the orthogonal approach combines the generation of many seamless chimeric IAA7 and IAA12 complexes with binding assays, MS-based proteomics, CD spectroscopy and in-silico protein docking and molecular dynamics simulation. The biochemical work appears to be high class work. However, we found some rather speculative conclusions from a structural point of view and a number of errors & oversights in the experimental procedures which potentially lead to inaccuracies in the study. Mindful of the impact of the story, we recommend major revisions and detail the specific points below in hopes of providing constructive points aiming to improve the manuscript and making it suitable for publication.

Major points

[1] In general the manuscript is not easy to read for readers lacking a solid background on the topic. We feel this is important given that the scope of the journal is to address a wide audience. To improve this, the abstract and introduction should be more general and with the basic concepts better explained without the need of reading all references cited. A major point here is to use less acronyms, which makes the text very difficult to follow. Additionally, the authors should put efforts in explaining why this work, which is primarily important for plant biologists, is also important to a more general audience. This can potentially be achieved by highlighting which technical/biological novelties are presented.

[2] Inspection of the repository containing the raw data of the cross-linking experiments appears to indicate a peculiar weakness in the data analysis of the raw data. The repository provides a FASTA file containing only the 4 proteins in the complex. If really used, this is directly counter to the recommendations of the Sinz laboratory, who have promoted the use of cleavable cross-linking reagents for many years to support the analysis of raw data with large(r) FASTA files. In addition, such low numbers of proteins will make FDR controlling the identifications at the indicated level impossible – although this situation will at least partly be rescued by only selecting reproducible cross-links. The authors should re-analyze the data with larger FASTA files to make the data analysis more secure.

[3] The number of cross-linked peptides seems very low as IAA7 and IAA12 targets contain many LYS residues and respectively 3 and 6 were found ubiquitinated. Such low density is counter-productive to the structural modeling steps the authors apply based on this data. To capture a higher density of cross-linked peptides, the authors can consider to employ pre-fractionation for their cross-linking reagent of choice for the samples after digestion to enrich for the cross-linked peptides and measure multiple fractions (as opposed to the single 'shot' currently analyzed). This obviously will depend on the amount of input material the authors will be able to generate, but will greatly assist in the modelling steps and increase the confidence in the results.

[4] Line 719: it is not clear why the authors choose to use cross-links present in 2/3 or 3/4 replicates and not to be consistent in this parameter – please comment on this. Additionally, it is currently impossible to assess which crosslinks were present in which replicate as the excel file in the PRIDE repository is corrupt (likely a result of a mistake in the data transfer). To improve on this point, apart from the need of updating it in the PRIDE repository to a non-corrupt version, the complete table containing the information and sequences of the crosslinks in the various replicates should be included in the supplementary material.

[5] Line 713: the cross-linkable residues set up in the search included LYS, SER, THR and TYR. To us, it is not clear how reliable these hits are, given that the preferable residues should be LYS-LYS, and what inclusion does to the false positive rates (a point for example apparently not researched in the recent paper Beveridge et al currently available in bioRxiv). The authors should reanalyze the data with LYS-LYS only and comment on the effect. Additionally, as the table containing the identifications in the PRIDE repository appears to be corrupt it is currently impossible for us to estimate the validity of the results (see also point 4).

[6] While inspecting the raw MS data for the XL-MS experiment it became clear that different instrument and acquisition settings were used. For instance, at least one of the samples (IAA12 rep.1, minIAA, plusT1) was acquired on a Q Exactive Plus with an MS2 resolution of 70'000 with stepped collision energy, while replicate 4 of the same sample was acquired on an Orbitrap Fusion with an MS2 resolution of 15'000 without stepped HCD collision energy. The above mentioned parameters are different from the one stated in the methods section of the manuscript. The authors should provide the correct settings in the manuscript and provide a rationale for the differences and/or potential effect.

[7] Line 728: "... we used distances from intramolecular cross-links of known distance (see SUPP file docking parameters)". The HADDOCK parameters are in fact not provided and it is also not clear which cross-links were used in which step, making it impossible to reproduce the result (Supplementary Table 3 contains solely information about the models generated by HADDOCK). Additionally, with "known distances" do the authors refer to the spacer arm length of the linker or the length of the side-chains added to the spacer arm length? In extension to this, are for example the detected Tyr then also adjusted with a different side-chain length? Many open questions which definitely can and should be answered by including a table with these details.

[8] Line 735: "Using the same restraints, the possible conformational docking space of the PB1 domains was searched and visualized using DisVis...". As stated by the authors who developed DisVis (ref 44 of this manuscript, doi: 10.1093/bioinformatics/btv333), DisVis is a "... powerful aid in detecting the presence of false-positive restraints". This means the tool should be used prior to the docking steps to select the set of distance restraints that define a specific interaction interface for optimal performance (also recommended in all their workshops). Supplementary Figure 11 appears to indicate the opposite in that both tools were run in parallel. In case this is true, the authors should re-run the docking procedures informed by DisVis. See point 7 on how to provide further details.

[9] Line 333: "The location of the clusters on two opposing surfaces of TIR1 334 suggests a rather extended fold of the AUX/IAA protein when bound to TIR1". How do the authors infer this extended fold of the protein if most of the crosslinks link the N-terminus to the C-terminus (see circus plots in Figure 4c and d)? We are wondering if cross-linking the recombinant expressed proteins with varying disordered tails would not be an excellent system to demonstrate whether the terminal binding remains intact – and if not whether this is the reason behind decreased binding.

[10] In extension to point 9, it seems strange that no extensive cross-linking is found on or in the neighborhood of the degron binding domain as this appears to be the main interaction point and the most stable point between the two proteins – which should lead to availability of many cross-linking options. The authors should comment on likely reasons to this, for example are no LYS-LYS pairs available in the interfaces between the proteins in the final model or other reasons.

[11] Figure 7: It is not clear how the authors came to this model. The PB1 domain, according to the reproducible cross-links considered, is linked to the K226 in the position "cluster 1" for the IAA7 (Figure 4c), while the KR motif at the N-terminus is linked to the K67 (not defined as a cluster but close to the ASK1 domain (Supplementary Fig. 9)). On the contrary, the PB1 domain of IAA12 is linked to K67 which is not compatible with the proposed model. Do the authors expect different sites of interaction for the PB1 domains of IAA7 with respect to IAA12?

Minor points

[12] Line 726: "...description how to prepare pdb files and incorporated distance restraint can be found elsewhere." Please provide a reference.

[13] Line 722: The comparative models of IAA7 and IAA12 PB1 Domains were created, but the PDB files are not available. Their inclusion we feel is important to allow readers to assess the claimed structural interactions based on the crosslinks found. Please provide these PDBs and possibly also the starting models used for the molecular dynamics simulation.

[14] Figure 5: The title specifies: ".. Cross-link-based docking.." and "... via HADDOCK...". The structures presented however are not docked PDBs, which is an HADDOCK output, but rather the possible conformations space occupied by the PB1 domain, which is an output of DisVis. Please here provide the proper docked structures or correct the figure caption.

[15] Line 327: "... in the absence of auxin, we only observed only a few inter-protein and similar intra-protein cross-links...". It is unclear to us from the provided figures 4 and sup 10 that this is the case. The authors should clarify this in more detail what the difference is.

Response to reviewers and additional information:

For access to datasets in PRIDE repository use PX reviewer account:

1) For ubiquitylation data:

Username: reviewer85637@ebi.ac.uk

Password: YXTsqTWX

Project accession: PXD015392

2) For XL-MS data:

Username: reviewer06502@ebi.ac.uk

Password: F3Tp3EKs

Project accession: PXD015285

Reviewer #1 (Remarks to the Author):

This manuscript combined an impressive array of methods to characterize the interaction between two AUX/IAA proteins with TIR1 in the context of auxin sensing. The authors identified new binding interfaces between AUX/IAA and TIR1 that were not present in the crystal structures in the 2007 Nature paper. Crosslinking MS experiments and mutagenesis data nicely corroborate to support the importance of the new binding sites, which allowed the authors to propose a structural model of the complex. The biochemical data are very solid and the findings are highly novel. These findings provided structural basis for understanding the differential auxin binding of the co-receptor complexes with different AUX/IAA components. However, a major caveat of the manuscript is that the paper solely relied on yeast two hybrid binding assays and in vitro data. Therefore, the manuscript would be much strengthened if the authors can show that the identified new interfaces actually affect AUX/IAA degradation in response to auxin in cells. For example, the mutations of TIR1 (D170K and S172A) t dramatically abolished binding of AUX/IAA in Y2H experiment, do they affect AUX/IAA degradation at all ? Same questions could be asked for the AUX/IAA chimera or deletion mutants. Adding the functional data would make the paper more relevant to plant genetic studies.

We thank the reviewer for the positive comments, and agreed that it was imperative to have *in vivo* evidence for the relevance of the newly identified TIR1· AUX/IAA interaction interfaces. We have therefore gone beyond the reviewer's suggestion, and obtained *in planta* data that substantiates our structural proteomics and biochemical approaches. We generated transgenic lines expressing TIR1 variants that contained mutations in residues in clusters 1 and 2 that, out of our in silico and in vitro data, we deem key for TIR1·AUX/IAA coreceptor formation. Further, we assessed whether these TIR1 mutant variants were capable of restoring the auxin resistant phenotype of *tir1-1* mutant plants. After an extensive analysis of hundreds of individual T1 lines, we determined that indeed single mutations of specific sites affected TIR1 function, as the transgenes did not complement resistance upon exogenously application of auxin (2,4-D and IAA) in a root elongation assays. We kindly invite the reviewer to look into this new data set that we have included as new Figure 7. Accordingly, we have expanded our model figure, which is now Figure 8.

More specific comments are below:

Fig 1C, disorder/beta sheet/alpha helical labels are difficult to distinguish, using a solid line ?

Yes, we absolutely agree and have now improved the visibility of the lines and the corresponding legend in the Figure.

Fig. 2b-2d, the authors need more repeats to generate statistics of the Kd values to draw conclusion about whether the different mutants indeed bind auxin with different affinity. This kind of measurement usually has a high variations. Also again, the curves are difficult to distinguish, please use different colors.

We appreciate this comment and agree in general. We generated additional replica for the chimera constructs with high errors (12-12-12-7-12 and 12-12-12-12-7), and included published Kd values for the wildtype proteins in Fig 2c. Please note every single point in the saturation binding curves comprise technical triplicates and 2-3 independent experiments. We have now soften the language in the corresponding part of the manuscript. More elaborated statistics were not included due to n=2 in some samples.

Supplementary Fig 1: would be nice to label the position of the degron in each panel.

Yes, this was indeed a good idea for better orientation of the reader. Supplementary Fig 1 was changed accordingly.

Fig. 1C, since IAA7 and IAA17 are mostly disordered, one would expect that it would be hard to purify the proteins. A coomassie staining gel of the purified proteins would be required to show the purity/integrity of the proteins. In addition, since the proteins are purified from E. coli, one cannot rule out that the prokaryote system might not fold the protein properly. Have the authors tried AUX/IAA expression in insect cells as TIR1-ASK1?

We could not find a correlation between intrinsic disorder and difficulties for heterologous expression/purification besides higher accessibility to proteolysis. In our case, we generated highly pure proteins, as one can observe in Supplementary Figure 3 in the coomassie-stained gels. For further experiments, we used proteins from only the early purest fractions. As for the second question of the reviewer, AUX/IAAs have been purified from E.coli by us and other groups for many years for numerous experiments, and we corroborate AUX/IAA activity, which indicates proper integrity, from e.g. auxin binding and ubiquitylation assays. Therefore, we do not see any need to switch the heterologous expression system to a less cost-efficient one.

Page 11 line 191, there is an extra "12" that needs to be removed.

this has been corrected

Page 12 line 205, Should be "Supplementary Fig. 4d".

this has been corrected

Page 12 line 215 Should be “Taken together”
this has been corrected

Supplementary Fig 8b, are these quantifications from several independent experiments, could the authors put p-values on the figure to indicate statistical significance. This is even more relevant given that in Figure 3, it really difficult to see the difference of the fluorescence intensity just by eyes, for example between 77777 and 777127. Also the orders of the samples between Supplementary Fig 8 and Figure 3 are different. Please make sure about the labeling and stick to the same order.

We agree with the reviewer, and have therefore improved Supplementary Figure 8. The order of the samples now match Figure 3, and depicted values correspond to normalized means to denote differences in basal AUX/IAA ubiquitylation. Statistical analyses in this case are not informative, due to the intrinsic nature of the method -quantification of a ubiquitylation signal in gel-, which varies from experiment to experiment. Nevertheless, the ubiquitylation trend within replicates is consistent.

Fig. 5a-b, What are the restrains indicated in the x-axis? Why is the additional degron tail restraint only restricted the accessible conformations in the last column group?

We thank the reviewer for asking us to clarify the labelling. The restrains in the x-axis are now included in the Supplementary Table 2. Numbers in the x-axis means the number of restrains satisfying complexes generated by DisVis. By incorporating additional restrains, few models are satisfied enabling to come closer to describing the interaction models of ASK1·TIR1·AUX/IAA protein complexes. The degron tail restraint only allows satisfaction of interaction spaces between 20-27 Å for IAA7 and 25-35 Å for IAA12. This is a strict distance criterion compared to cross-linking data, which may satisfy distances between 8-34 Å. This is the reason inclusion of the additional degron tail restrains generates fewer satisfied models. See details in the expanded Methods section.

Reviewer #2 (Remarks to the Author):

Auxin is the most critical hormone in plant biology, driving all cell division and cell elongation events. Despite its great importance, its signal transduction pathway is seemingly simple, with a repression – depression paradigm, driven by auxin-mediated interactions between the SCFTIR1 complex and an Aux/IAA repressor, ruling auxin responses.

In this manuscript, the authors use a variety of approaches to resolve the topology of TIR1 contact points with IAA7 and IAA12, with a focus the C-terminal intrinsically disordered regions of these proteins. Overall, this work will provide important details to the auxin community that may also be relevant to interactions among other F-boxes with their targets.

We appreciate the reviewer recognizes the power of our findings and approaches to address and map protein-protein interactions beyond plant systems, and to unveil e.g. target recognition domains in ubiquitylation targets.

I have a few minor comments for improving this manuscript.

1- Line 264– It seems strange to me that the authors did not cite Gilkerson (2015) who showed by mutational analysis that Aux/IAA polyubiquitylation is flexible and can take place on many potential residues. At the very least, this should be discussed in the context of their work.

Yes, we recognize the omission. This occurred as we had to comply with the restrictions of Nat Comms regarding the number of citations we could include in the manuscript. We have now included the missing reference.

2- Line 346 – The authors make a strong statement about the numbers of assemblies detected for IAA12bm3 and IAA7bm3 and distribution of inter-protein crosslinks. Based on difficulties in identifying all peptides in mass spec, I am not comfortable with such a strong conclusion – the authors should soften their statements on this.

We understand the reservation of the reviewer in this regard. We would like to clarify, however, that the number of intra-protein XLs we detected along the whole sequence of AUX/IAA was the best indicator for very good sequence coverage. We included now Supplementary Files 1 and 2 showing the crosslinking identification, specifically the spectra number per unique XL, which corroborates our statements, and the reviewer can now access. We apologize for not having included this data in the first submission.

3- The theme of hugging and kissing and embracing throughout the discussion to describe TIR-Aux/IAA interactions seems inappropriate. I am guessing that a “hug” and a “kiss” are meant to depict different types of interactions, but it is difficult to tell. I would suggest using more specific, non-colloquial, language.

In the revised version of the manuscript we have changed the language explaining the proposed mechanism of TIR1-AUX/IAA engagement, as suggested by the reviewer.

Specific Comments on the Modeling and XL-MS sections

Reviewer #3 (Remarks to the Author):

Authors used interdisciplinary approach and combined an in silico analysis, biochemical analysis of recombinantly expressed proteins, the yeast two hybrid system (Y2H) for qualitative assessment of auxin co-receptor assembly as well as ubiquitylation dynamics of selected proteins (IAA7 and IAA12). Using spectroscopic (CD) and diffusional experiments in combination with recombinantly expressed proteins they found that N terminal part of these proteins is disordered with an extended structure in solution. Using Y2H approach of chimeric proteins they investigated impact of intrinsic disordered segments as well as of the compact PB1 domain of AUX/IAA (IAA7 and IAA12) proteins on their interaction with TIR1, auxin independent, as well as auxin-triggered.

They stated that they confirmed the postulated interdependency of the degron tail and PB1 domain, but I am not convinced since chimeric proteins without PB1 nicely bind to TIR1, also, there is no experimental data (or I overlooked them?) on the PB1 - TIR1 binding. They point to additive and separate effects of each, disordered degron tail and the PB1 domain, on auxin independent and auxin-triggered TIR1 interaction.

We thank the reviewer for her/his critical view, as it became clear to us that we needed to elaborate and clarify in the manuscript various concepts on auxin signaling many readers would be probably not familiar with. Over a decade of research on AUX/IAAs in Arabidopsis has evidenced their PB1 domain serves as homo- and hetero-dimerization domain for AUX/IAA-AUX/IAA and AUX/IAA-ARF interactions. In our dataset, besides computational binding, there is no evidence for TIR1-PB1 direct binding, unless the AUX/IAA PB1 domain is connected via a degron tail to the degron, which is the main driving force of TIR1-AUX/IAA interactions. Throughout the manuscript we point out, that TIR1-PB1 association is an additional fine-tuning mechanism, and that this fine-tuning might be modulated via PB1-degron tail dependencies in an AUX/IAA specific manner. The reviewer further pointed out, that “chimeric proteins without the PB1 nicely bind to TIR1”. This is only true for the Y2H assay, and, it is likely an artefact of the yeast system, as shown also by others (Moss et al Plant Phys 2015). An AUX/IAA carrying an intact PB1 domain potentially undergoes oligomerization in yeast, which diminishes the signal corresponding to the activation of the LacZ reporter construct. Delta PB1 domain versions of AUX/IAAs are then not hindered in those assays, and therefore TIR1-AUX/IAA delta PB1 interactions in yeast appear stronger than TIR1-AUX/IAA interactions. We have recognized the caveat of this approach, and understand that deletion constructs of AUX/IAAs in yeast are not informative, and have therefore removed these chimeric protein from the Y2H data set in Figure 2.

Furthermore, they postulated that AUX/IAA ubiquitylation favorably occurs at the solvent exposed regions in IAA7 and IAA12, when they are recruited by TIR1. Many assumption but without firm evidences.

In order for ubiquitylation, or for that matter any other PTM to occur on specific residues, those must be exposed. Sites at the N-termini of both IAA7 and IAA12, and degron tail of IAA12 undergo ubiquitylation, and those precise regions were identified in our XL experiments. XL spectra would not have been observed, unless these regions are solvent exposed. Furthermore, final structures refined after MD

simulations showed the ubiquitylated lysines in the IAA7 PB1 domain to be solvent exposed. Therefore, the overlap of our IVU and XL findings supports our arguments.

Based on the structural proteomics approach authors assumed AUX/IAAs IAA7 and IAA12 exhibit flexible conformations in solution (intra-protein cross-links), and adopt an extended fold when bound to TIR1. But the cross-linking reactions were performed for chimeric (PB1 missing domain) proteins, and to get more convincing assumption the data for the native proteins are needed.

We suspect this is a misunderstanding from the reviewer's part, as our XL data was acquired using the full length of TIR1 and AUX/IAA proteins. Thus, our arguments on AUX/IAA flexibility are not only based on crosslinks, but on CD and SEC data.

Y2H

I am not an expert, but from the data shown, it is hard to follow the statements given in the manuscript. For example:

1. The statement on IAA7 and IAA12 – TIR1 binding affinity ' Expression of the β -galactosidase reporter indicates stronger interaction of TIR1•IAA7 than TIR1•IAA12.' Is not correlated with Fig. 2a, since according to the data shown it seems that TIR1•IAA12 bind only at VERY high IAA conc. (> 100 μ M).

We recognize the confusion of the reviewer, and the difficulties one might encounter when assessing these different, but complementary approaches. On one hand, Y2H protein-protein interaction data (TIR1-AUX/IAAs) (strong vs. weak interactions, according to reporter expression) is an expression of auxin receptor formation. On the other hand, radioligand binding assays evidence the ability of TIR1-AUX/IAA complexes to bind auxin, and the affinity is expressed as Kd values (Kd high value (high nM and beyond) = low auxin binding affinity, Kd low value (low nM range)= high binding affinity). We invite the reviewer to reassess the statements in the manuscript, as they correctly describe the interactions and binding events.

2. Author wrote that they docked homology-modeled PB1 domains of Arabidopsis IAA7 and IAA12 to the ASK1-TIR1 complex, applying distance restraints based on the cross-linking data. The mentioned distance restraints are related to the regions out of PB1, and it is not clear how authors modeled 3D structure of these regions?

We thank the reviewer for the useful comment. We have updated the Methods part of the manuscript to describe better the modelling procedures. In brief, we modelled only the folded PB1 domain of IAA7 and IAA12 proteins, and not the flexible rest of the proteins, which is currently an impossible task for any structure prediction algorithm. In addition, we used the crystallographically-determined degron of IAA7 bound to TIR1, without modelling the AUX/IAAs further. We have now included the alignment files, the input crystal structures and the models generated by MODELLER in Supplementary File: modeller_files.tgz.

As for the docking, we have only used 3 cross-links for IAA7 and the two cross-links for IAA12, which connect TIR1 to the PB1 domains of the AUX/IAAs. The novelty and the significant reduction in the derived models is due to the fact we used a new type of restraint in both DisVis calculations and HADDOCK docking, that of the minimum and maximum allowed distance between the C-ter of the

degron and the N-ter of the PB1 domain (see new Supplementary Table 2). Therefore, no cross-links outside the structurally defined regions were used for the docking.

Ubiquitylation

3. The statement: 'IAA12~ubiquitin conjugates were much less abundant than IAA7 after 30 min incubation' cannot be derived from the data represented in Fig3a. From Fig. 3a it seems that they (IAA7 and IAA12) are similarly ubiquitinated.

We thank the review for pointing out this and agree. We have corrected accordingly our statements in the new version of the manuscript.

Cross-linking

How long is the cross linker used? Are there data on its influence on the protein conformation?

The crosslinker we used is the previously published DSBU/BuUrBu crosslinker, which has a total spacer arm length of 12.5 Å. Reasonable detected C α -C α distances for lysine-reactive crosslinkers like DSBU range between 7-30 Å with maximum distances up to 35 Å. Since the beginning of cross-linking/MS (XL-MS) in 2000, a large body of publications has proven that XL-MS does not induce artificial conformations in proteins if the conditions are carefully controlled. One of the main advantages of XL-MS is it can capture different protein conformations overlying each other, as well as transient protein-protein interactions. Over-cross-linking that might result in artificial protein conformations has been avoided by performing functional assays of the proteins under investigation.

Figure 6:

4. Figs 6a and 6b - From the ΔG graphs it seems that the affinity of PB1(IAA7) for TIR1 is lower than that of PB1(IAA12) – TIR1, while according to the experiments IAA7 binds to TIR1 with higher affinity than IAA12

We agree with what the reviewer stated here, but the affinities (for auxin binding) are coming from full length proteins, and aim specifically for the binding of auxin, while the MD simulations solely predict the PB1 domain interaction with TIR1. Both experiments need to be treated individually, and even though the PB1 domain might interact stronger with TIR1 (based on the MDs), the auxin-driven interaction relies mainly on the core degron as published in Tan et al. Nature 2007 and Calderon Villalobos et al Nature Chemical Biology 2012.

5. The following statement: 'Mutations R156E, as well as S201A, and S205A either abolished or drastically impaired basal TIR1•IAA7 and auxin-driven TIR1•IAA7 and TIR1•IAA12 associations, without affecting ASK1•TIR1 assembly.' could not be deduced from Fig 6 e. According to the figure the statement is only partly true for R156E mutants of IAA7 at low auxin conc. (0.1 and 1 μ M), but for IAA12, as well as for IAA7 at higher IAA conc, there is no significant difference with respect to the WT proteins.6.

We thank the reviewer on this comment, as it made us reassess the Y2H data in Figure 6, and rephrase the previously unclear statements in the manuscripts. We have performed these Y2H experiments again with a selection of TIR1 mutants, and addressed their ability to interact with ASK1 independently of

auxin, or in an auxin-dependent manner with AUX/IAA. The new Y2H data can be found in an updated Figure 6. Our *in silico* and *in vitro* data has now be substantiated *in vivo*, and the readers can find our newly acquired data in Figure 7.

Fig2b, the affinity curve for IAA/bm3 have not reached the saturation (plateau)

The reviewer is correct. The initial IAA concentrations for IAA7 and IAA7BM3 were based on previous observations (Calderon Villalobos Nat Chem Biol 2012) and unfortunately did not lead to a full plateau. Since we obtained similar K_d s as published, we are confident that our new calculated K_d s are reliable. Data points for the chimeric 7-7-7-12-7 and 7-7-7-7-12 proteins were obtained for IAA concentrations up to 1000 nM. Thanks to the reviewer's comment, we checked the influence of the last 2 concentrations (500 and 1000 nM) on the K_d values of those, and found only minor changes, which don't affect the overall outcome of the analysis.

The computational methods used in this manuscript is really strange:

1. Authors performed homology modelling of IAA7 and IAA12 PB1 domain, but what is with the other regions of the protein (DI, linker, degron tail?). How many and what structures were used as templates? What was the quality of the obtained models, how big is the 3D structure?

We would like to thank the reviewer for asking us to be more detailed. We have updated the methods part, and have added a Supplementary File: `modeller_files.tgz`, which includes input files, alignment files, as well as the generated models for the PB1 domains of AUX/IAA proteins. In brief, as mentioned in Y2H point #2, we have only modelled the PB1 domains of the AUX/IAA proteins, and not the flexible regions. Overall the models are of good quality, alignments are very straightforward and Modeller includes stereochemical checks minimizing the inter- and intra-molecular contacts. The structures are ~35 Å in diameter, the PB1 domain of IAA7 contains 117 residues and the PB1 domain of IAA12 contains 124 residues. The reviewer can now access the PDB files, and judge the quality of the generated models.

2. Authors wrote that 'how to incorporate distance restraint' as input for HADDOCK can be found elsewhere (they did not give a reference), but these restraints are related to regions out of PB1 while they have homology model only of PB1 domain. How they combined distance restraint data with non existing 3D structure for the purpose of docking?

We have now included the appropriate references on how to include distance restraints in HADDOCK, and updated the Methods section of the manuscript. For the purpose of docking, we have created Supplementary Table 2, where the restraints used, are clearly defined. In brief, we used the TIR1·ASK1·degron(IAA7) structure (PDB: 2P1Q) and the PB1 domain of IAA7 to produce higher order assembly, which satisfies the 3 cross-links mapping to the ordered domains mentioned above. In addition, the C-ter of the degron and the N-ter of the PB1 domain, because they are in the same protein, are separated by a distance, which can be calculated as in the Hamdi et al. 2017 publication. Therefore, with these 4 restraints, we are able to generate interaction models of the ordered domains of the proteins. We repeated the same method for deriving higher-order assemblies for the TIR1·ASK1·degron structure of IAA12, and the PB1 domain of IAA12. For the reviewer and the readers to have access to these calculations, we have included the parameters of the docking in a tar file (`haddock_files.tgz`) as a separate Supplementary File.

3. Molecular modelling procedure: Structures (PB1(IAA7)-TIR and PB18(IAA12)-TIR) were optimized in MOE (using AMBER10 ff),

But there is no reference. As far as I know AMBER10 ff does not exist! Further on, the simulations were performed in GROMACS with AMBER99SB-ILDN force-field (the energy minimization and MD simulations should be performed with the same ff).

We thank the reviewer for the useful comment. We have now included the reference of Case et al. 2008 for Amber10ff. A first, rather less robust, energy minimization was indeed performed in MOE with AMBER ff10. However, we performed an additional energy minimization using the AMBER99SB-ILDN force field, along with position-restrained equilibration (as stated in the Methods) in GROMACS 4.6.5. A more detailed section of the Supplementary MD methods has now be included for clarity.

Authors states that 'electro-neutrality was guaranteed with a NaCl concentration of 0.2 mol/L'. What was net charge of the system! Electro-neutrality can not be achieved by adding the same amount of Na+ and Cl-, but, by adding certain number of either Na+, or Cl- ions, depending on the charge of the system.

Electro-neutrality in the manuscript states that the system has a net charge of zero with a NaCl concentration of 0.2M, it does not imply that the amount of Na and Cl ions is the same. This sentence was rewritten in the manuscript for better understanding.

Nothing about time step, ensemble, barostat, thermostat ...

The reviewer is right regarding this point. We are sorry these parameters were not originally included in the manuscript due to space constraints. We have now added an extended version of the MD methods (Supplementary Methods), and a Supplementary file (MD.tgz), including the used MDP files (for energy minimization, equilibration and productive simulation), as well as the t=0ns PDB file from each simulation run.

Finally, the binding free energies were calculated by MMPBSA.py from Amber18 package. Why, not in Gromacs, since it was used for MD simulations, and it has his own MMPBSA tool. Why usage of so many different program packages (and ff), when everything could be done within either GROMACS the AMBER18 package?

GROMACS and AMBER have a nice compatibility, the MMPBSA.py tool implemented in AMBER is easy to use and our system is well adapted to execute both programs and handle large amounts of data (for instance each time point of a simulation).

Figures 6 a & b: Authors wrote: 'The accumulated mean value of ΔG_{eff} were obtained every 10 ps' What does mean 'accumulated'? It should be the 'mean value of ΔG_{eff} ' (or running average). What was the frequency of sampling?

We understand how this may be confusing for the readers, and thank the reviewer for asking us to be more detailed. The MMPBSA.py calculates the ΔG_{eff} every 10 ps, so we have 2 000 ΔG_{eff} values for the whole 20 ns simulation. We calculated then the cumulative mean (a recursive method) for each of those 2000 values. We computed the (accumulated) mean for each position by summing over all

previous values and dividing by their number. The calculation procedure of the accumulated mean is now included in the new Methods section.

In general the sentences are too long, awkward and imprecise, for example:

' The residency time of an AUX/IAA target on TIR1, when assembled in an SCF-complex, propels processivity of AUX/IAA ubiquitylation...'

(How time can propels something, maybe correlate OR define ...)

We acknowledge this might have been the case in certain stances along the previous version of the manuscript. We have addressed and corrected the stances accordingly in the new version of the manuscript.

' IDR flanking the degron provide high flexibility and an extended fold to AUX/IAAs'

Should be 'ensure' (or enable..)

this has been corrected

' ΔG_{eff} values over 20 ns identifying and refining stable TIR1•AUX/IAA PB1 complexes'...

Identifying is OK, but the calculated ΔG_{eff} values do not refine structures!

this has been corrected

Also, some citations are (in my opinion) not adequate, for example for the statement 'Under low auxin concentrations, AUX/IAAs repress type A AUXIN 53RESPONSE FACTORS (ARF) transcription factors via physical heterotypic interactions through their type I/II Phox/Bem1p (PB1) domain'

authors cite: Liscum, E. & Reed, J. Plant Mol Biol (2002) 49: 387,

instead of

Tiwari S.B., Hagen G., Guilfoyle T.J. (2004). Plant Cell 16: 533–543

OR

Tom J. Guilfoyle Plant Cell. 2015 Jan; 27(1): 33–43.

We have made the appropriated corrections. We recognize that the limitations in the number of references (70), we can include in the Nat Comms manuscript, makes difficult for us to acknowledge the tremendous amount of work that has contributed to a better understanding of the auxin field in the last two decades. We apologize for any unintentional omission.

Reviewer #4 (Remarks to the Author):

The paper by Niemeyer et al presents an interesting hypothesis on the role of intrinsically disordered regions in the auxin-mediated interaction between INDOLE-3-ACETIC ACID proteins (transcriptional repressors) and F-BOX PROTEINS TRANSPORT 57 INHIBITOR RESPONSE 1 (TIR1)/AUXIN SIGNALING F-BOX 1-5 (AFB1-5). The authors claim as their main conclusion that the intrinsic flexibility of AUX/IAAs ultimately affects the ubiquitination system, and in turn proteasome recruitment. With this angle, we believe the manuscript has considerable impact due to the crucial role of the ubiquitin-proteasome system which, in combination with the effect of Auxin concentration, is crucial to regulating plant growth.

The work represents additionally a serious effort in its experimental setup, in which the orthogonal approach combines the generation of many seamless chimeric IAA7 and IAA12 complexes with binding assays, MS-based proteomics, CD spectroscopy and in-silico protein docking and molecular dynamics simulation. The biochemical work appears to be high class work. However, we found some rather speculative conclusions from a structural point of view and a number of errors & oversights in the experimental procedures which potentially lead to inaccuracies in the study.

Mindful of the impact of the story, we recommend major revisions and detail the specific points below in hopes of providing constructive points aiming to improve the manuscript and making it suitable for publication.

We thank the reviewer for highlighting the quality of our biochemical work, and for recognizing the scope of our pioneering interdisciplinary approach.

Major points:

[1] In general the manuscript is not easy to read for readers lacking a solid background on the topic. We feel this is important given that the scope of the journal is to address a wide audience. To improve this, the abstract and introduction should be more general and with the basic concepts better explained without the need of reading all references cited. A major point here is to use less acronyms, which makes the text very difficult to follow. Additionally, the authors should put efforts in explaining why this work, which is primarily important for plant biologists, is also important to a more general audience. This can potentially be achieved by highlighting which technical/biological novelties are presented.

We absolutely understand the point raised by the reviewer, and have made amendments in the text to introduce the topic to non-specialists.

[2] Inspection of the repository containing the raw data of the cross-linking experiments appears to indicate a peculiar weakness in the data analysis of the raw data. The repository provides a FASTA file containing only the 4 proteins in the complex. If really used, this is directly counter to the recommendations of the Sinz laboratory, who have promoted the use of cleavable cross-linking reagents for many years to support the analysis of raw data with large(r) FASTA files. In addition, such low numbers of proteins will make FDR controlling the identifications at the indicated level impossible – although this situation will at least partly be rescued by only selecting reproducible cross-links. The authors should re-analyze the data with larger FASTA files to make the data analysis more secure.

We are thankful for the reviewer's comment, as this was an unintended omission. We have therefore revisited the data in the repository. Our analyses included always larger fasta files comprising the "common Repository of Adventitious Proteins" (cRAP) database, which contains more than 100 general lab contaminants, which we partially also identified in our dataset. Those are not directly included in the uploaded fasta file, but defined by the MeroX parameters/settings.

[3] The number of cross-linked peptides seems very low as IAA7 and IAA12 targets contain many LYS residues and respectively 3 and 6 were found ubiquitinated. Such low density is counter-productive to the structural modeling steps the authors apply based on this data. To capture a higher density of cross-linked peptides, the authors can consider to employ pre-fractionation for their cross-linking reagent of choice for the samples after digestion to enrich for the cross-linked peptides and measure multiple fractions (as opposed to the single 'shot' currently analyzed). This obviously will depend on the amount of input material the authors will be able to generate, but will greatly assist in the modelling steps and increase the confidence in the results.

We thank the reviewer for the helpful comment and in general agree, that more crosslinks would be helpful for the later modeling steps. However, we wonder how the reviewer made the connection to the ubiquitylation as both experiments were carried out completely separately (and with fresh, independently expressed and purified proteins). Therefore, ubiquitylation is not interfering with the crosslinking experiments. To further clarify, we incorporated now two Supplementary Files 1 and 2, which show the number of unique crosslinks, and the number of spectra identifying each. For intraprotein XLs in the combined IAA7 datasets (12 reactions in total) we reach up to 568 spectra for one unique XL and 16.25 spectra per unique XL on average. For intraprotein XLs in the combined IAA12 datasets (16 reactions in total), we reached up to 552 spectra for one unique XL and 28.24 spectra per unique XL on average. For TIR1-AUX/IAA interprotein crosslinks those numbers are greatly reduced (IAA7: max=32, average=5.59; IAA12: max 35, average= 3.68). The main limitation in those experiments is indeed the amount of recombinant in-insect cells -expressed TIR1 available, we are confident though, our data set and the stringent analysis suffice.

[4] Line 719: it is not clear why the authors choose to use cross-links present in 2/3 or 3/4 replicates and not to be consistent in this parameter – please comment on this. Additionally, it is currently impossible to assess which crosslinks were present in which replicate as the excel file in the PRIDE repository is corrupt (likely a result of a mistake in the data transfer). To improve on this point, apart from the need of updating it in the PRIDE repository to a non-corrupt version, the complete table containing the information and sequences of the crosslinks in the various replicates should be included in the supplementary material.

We apologize for the data format deposited in PRIDE repository, and the inconveniences this might have caused while trying to read the XL files. The PRIDE deposited file is actually not corrupt, as the identification was done directly in MeroX, and no separate identification file was generated. We rather used the wrong file format in the first submission. Unfortunately, the PRIDE repository cannot deal with MeroX-derived XL data (zhrm files), and we have therefore initially uploaded the result files (zhrm) renamed (csv). Also, we now included a Disclaimer/Readme file in the PRIDE deposited data, which

describes the way to read out the data. In general the single or the combined results files (.zhrm) can be directly opened using MeroX and spectra related to each crosslink can be inspected.

Regarding the experimental approach, we have increased the number of replica in case of IAA12, as the crosslinks we found were less consistent in the first analyses. We improved the Methods section, and commented accordingly in the Results section. We added Supplementary Files 1-2 depicting the crosslinks identified and specified the replica.

[5] Line 713: the cross-linkable residues set up in the search included LYS, SER, THR and TYR. To us, it is not clear how reliable these hits are, given that the preferable residues should be LYS-LYS, and what inclusion does to the false positive rates (a point for example apparently not researched in the recent paper Beveridge et al currently available in bioRxiv). The authors should reanalyze the data with LYS-LYS only and comment on the effect. Additionally, as the table containing the identifications in the PRIDE repository appears to be corrupt it is currently impossible for us to estimate the validity of the results (see also point 4).

We appreciate the reviewers comment, but disagree in this point, as it is clearly stated in a community-wide publication (<https://doi.org/10.1021/acs.analchem.9b00658>) (where this expert reviewer is probably a co-author), that SER, THR, TYR crosslinks can be valuable for samples containing a low number of proteins, and should only be excluded for complex samples, such as full lysate and proteome-wide analyses. We therefore did not re-analyze the whole dataset using Lys-Lys crosslinks only. As mentioned one point above, we included now additional Supplementary Files 1-2 with the identified crosslinks and added a Readme/Disclaimer file, which facilitates the accessibility of the deposited data.

[6] While inspecting the raw MS data for the XL-MS experiment it became clear that different instrument and acquisition settings were used. For instance, at least one of the samples (IAA12 rep.1, miniIAA, plusT1) was acquired on a Q Exactive Plus with an MS2 resolution of 70'000 with stepped collision energy, while replicate 4 of the same sample was acquired on an Orbitrap Fusion with an MS2 resolution of 15'000 without stepped HCD collision energy. The above mentioned parameters are different from the one stated in the methods section of the manuscript. The authors should provide the correct settings in the manuscript and provide a rationale for the differences and/or potential effect.

We are thankful for the reviewer's thoughtful data check. We indeed found multiple files (8 from 38), which were run on different/unwanted settings due to an honest mistake. We omitted those data sets from the revised manuscript, and repeated the experiments with newly generated protein samples, and repeated the full analysis with the newest MeroX version. All results, figures and files were updated accordingly. Our findings were reproducible, and our new results were consistent with our original XL data, with only minor differences in individual spectra, which did not change the overall outcome of our conclusions, but rather strengthen them.

[7] Line 728: "... we used distances from intramolecular cross-links of known distance (see SUPP file docking parameters)". The HADDOCK parameters are in fact not provided and it is also not clear which cross-links were used in which step, making it impossible to reproduce the result (Supplementary Table 3 contains solely information about the models generated by HADDOCK). Additionally, with "known distances" do the authors refer to the spacer arm length of the linker or the length of the side-chains

added to the spacer arm length? In extension to this, are for example the detected Tyr then also adjusted with a different side-chain length? Many open questions which definitely can and should be answered by including a table with these details.

We admit that our methods section, especially the computational part, needed to be more detailed. We updated the methods section accordingly, and provide now all parameters regarding the docking as a supplemental file (haddock_files.tgz), including the docking parameters. Furthermore, Supplemental Table 2 clearly specifies now which distance restraints were used and their origin. Furthermore, the known distances we refer to, are distances we could identify from an albumin contamination (was used as a passivation agent in ultrafiltration concentrators) within our samples and additional the published distance values which are observed for DSBU.

[8] Line 735: “Using the same restraints, the possible conformational docking space of the PB1 domains was searched and visualized using DisVis...”. As stated by the authors who developed DisVis (ref 44 of this manuscript, doi: 10.1093/bioinformatics/btv333), DisVis is a “... powerful aid in detecting the presence of false-positive restraints”. This means the tool should be used prior to the docking steps to select the set of distance restraints that define a specific interaction interface for optimal performance (also recommended in all their workshops). Supplementary Figure 11 appears to indicate the opposite in that both tools were run in parallel. In case this is true, the authors should re-run the docking procedures informed by DisVis. See point 7 on how to provide further details.

The reviewer raises an interesting point on how to use programs developed in the Bonvin group. Actually, one of the authors is a major developer of HADDOCK and he, in fact, performed these calculations. In brief, DisVis and HADDOCK are two independent programs, serving different purposes. In HADDOCK, atomic interaction models can be generated, given sufficient restraints; DisVis, in contrast, allows to visualize and quantify the information content of distance restraints between macromolecular complexes. Of course, results can be combined from the two softwares, but users must also be careful not to generate HADDOCK models that do not satisfy the cross-linking data, because the active residues that are reported by DisVis are a statistical average and are not always involved in the interaction. That was the reason that the two programs were used separately. The programs actually converged to similar solutions, given the fact that the restraints that were submitted to both, were identical.

We have followed the comments of the reviewer, extracted active residues from DisVis, and used them as input for a HADDOCK run with cross-links and DisVis active residues as distance restraints for both IAA7 and IAA12 docking runs. These two runs are now provided as Supplementary Files disvis_haddock.tgz, and parameter are provided so that the readers and the reviewer can reproduce the generated models. Overall, both runs resulted in clusters with high restraint violation energies as not all restraints generated from the extracted active residues can be satisfied at once. Nevertheless, at least two clusters could be identified that were similar to the starting structures used for MD simulations, and therefore the combination of DisVis and HADDOCK validated the previously generated models from HADDOCK. The superimposed models of the best clusters (Cluster 3 for IAA7 and Cluster 2 for IAA12) can be found in Supplementary Figure Review (see next), as well as a representation of all the new generated models from HADDOCK. Due to the high restraint violation energies obtained from the combination of DisVis and HADDOCK, this figure was not included in the main manuscript.

Supp. Fig. Review

Supplementary Figure Review| The combination of DisVis and HADDOCK validates the previously generated models from HADDOCK. Both HADDOCK runs (ASK1·TIR1·IAA7^{PB1} and ASK1·TIR1·IAA12^{PB1}) resulted in more clusters that differ from the previously generated models. Nevertheless, we identified a few clusters that were similar to the starting structure used for MD simulations. Different clusters of AUX/IAA (IAA12^{PB1} in dark blue and IAA7^{PB1} in dark orange spheres), obtained from the combination of DisVis and HADDOCK, are represented through their centers of mass. Superimposed models of the most similar cluster (transparent) to the MD starting structure (IAA7^{PB1} in light orange and IAA12^{PB1} in aquamarine). The TIR1 structure is shown as light pink cartoon representation.

[9] Line 333: “The location of the clusters on two opposing surfaces of TIR1 334 suggests a rather extended fold of the AUX/IAA protein when bound to TIR1”. How do the authors infer this extended fold of the protein if most of the crosslinks link the N-terminus to the C-terminus (see circus plots in Figure 4c and d)? We are wondering if cross-linking the recombinant expressed proteins with varying disordered tails would not be an excellent system to demonstrate whether the terminal binding remains intact – and if not whether this is the reason behind decreased binding.

Excellent comment from the reviewer. We infer this extended fold, as we see a clear tendency of crosslinks downstream of the degron (degron tail and PB1 domain) towards cluster 1 while crosslinks upstream of the degron (N-ter) rather connect to cluster 2. Furthermore, the orientation of degron itself hints towards this direction, and the extended fold of the unbound AUX/IAAs in solution described in Figure 1. The statement mentioned was written as a hypothesis, because we are aware of the rather low resolution of the XL-MS approach. Indeed, modifying the length of the N-terminal would be of great interest and the influence of the length of different segments adjacent to the degron was partially investigated in other publications (Moss et al Plant Phys 2015, Guseman et al Development 2015) from a biology point of view. A detailed investigation on the dynamics of the N-terminus of AUX/IAAs is, in our opinion, out of the scope of this manuscript, and should be addressed in future studies.

[10] In extension to point 9, it seems strange that no extensive cross-linking is found on or in the neighborhood of the degron binding domain as this appears to be the main interaction point and the most stable point between the two proteins – which should lead to availability of many cross-linking options. The authors should comment on likely reasons to this, for example are no LYS-LYS pairs available in the interfaces between the proteins in the final model or other reasons.

The reviewer brings up a point, we have not considered so far. The distribution of lysine residues on TIR1, and in the degron flanking regions might be indeed the reason we find no additional crosslinks in those regions. One can also note from our data, that crosslinks in those regions might be favored as the number of spectra identifying those are higher (32 spectra for IAA7_K99-XL-TIR1_K217, 21 spectra for IAA7-S116-XL-TIR1_K226). Overall, the distribution of lysines on TIR1 is quite equal but the one within the AUX/IAA sequence is not. We have now addressed this point in the main text of the manuscript. Additionally, the reviewer and the reader can now access the XL-MS data in more detail in the Supplementary Files 1 and 2.

[11] Figure 7: It is not clear how the authors came to this model. The PB1 domain, according to the reproducible cross-links considered, is linked to the K226 in the position “cluster 1” for the IAA7 (Figure 4c), while the KR motif at the N-terminus is linked to the K67 (not defined as a cluster but close to the ASK1 domain (Supplementary Fig. 9). On the contrary, the PB1 domain of IAA12 is linked to K67 which is not compatible with the proposed model. Do the authors expect different sites of interaction for the PB1 domains of IAA7 with respect to IAA12?

We thank the reviewer for the comment, and want to clarify that the model seeks to highlight the that AUX/IAA extended fold, and that regions outside of the degron, such as the PB1 domain take part in auxin receptor formation. We agree, that the depiction might be misleading, as there are rather multiple

possible slightly different positions for the PB1 domain. We thought overlapping depictions of multiple PB1 domains would unnecessary complicate the model. Still, the regions where the PB1 domains of IAA7 and IAA12 are contacting TIR1 are overlapping, and K67 in TIR1 contacts the slightly extended very C-terminus of the IAA12 PB1 domain. This crosslink is at the upper distance limit permitted in the model generated after 20 ns MD simulation. In general, the positioning of the IAA12 PB1 might be more flexible compared to the IAA7 PB1 domain. All in all, yes we think the position of both PB1 domains is at least slightly different and might translate in differential influence on complex formation/auxin binding. We have improved the depiction in the model and highlighted the TIR1 residues engaging with regions outside of the degron of AUX/IAAs, which were specifically identified to be relevant in planta.

Minor points:

[12] Line 726: "...description how to prepare pdb files and incorporated distance restraint can be found elsewhere." Please provide a reference.

The reference has been provided in the new version of the manuscript.

[13] Line 722: The comparative models of IAA7 and IAA12 PB1 Domains were created, but the PDB files are not available. Their inclusion we feel is important to allow readers to assess the claimed structural interactions based on the crosslinks found. Please provide these PDBs and possibly also the starting models used for the molecular dynamics simulation.

We agree and have uploaded all models according to the generation. The comparative models can be found in the Supplementary file `modeller_files.tgz`.

[14] Figure 5: The title specifies: "... Cross-link-based docking.." and "... via HADDOCK...". The structures presented however are not docked PDBs, which is an HADDOCK output, but rather the possible conformations space occupied by the PB1 domain, which is an output of DisVis. Please here provide the proper docked structures or correct the figure caption.

We agree and changed the figure caption accordingly. Furthermore, we now specified the differences more clearly.

[15] Line 327: "... in the absence of auxin, we only observed only a few inter-protein and similar intra-protein cross-links...". It is unclear to us from the provided figures 4 and sup 10 that this is the case. The authors should clarify this in more detail what the difference is.

We value the reviewers comment, and the exact numbers of spectra identifying interprotein XLs in TIR1·AUX/IAA complexes are now compared in the new Supplementary files. The spectra found in the absence of auxin are about 10% of those found in the presence of auxin.

Reviewers' comments:

Reviewer #4 (Remarks to the Author):

The authors have performed an impressive array of work for this rebuttal and I commend them on their efforts. The readability of the manuscript has improved significantly and most of the requests for clarification about the structural modelling addressed. In addition, the MS experiment are now replicated and the instrumentation used is now consistent across the different runs.

In my opinion there remain however three small points related to the modelling procedures that need to be addressed before publication, as I believe they are important to the work.

[1] In the file "haddock_files.tar\haddock_files\IAA07\No_disorder_restraint\", the PDB cluster 1_1 is missing. In the summary of the run it is indicated as present. It is an important point to clarify as this cluster is the most significant (124 of the 175 structures) and the first structure is usually the highest scoring one, so it should be included or its absence explained.

[2] The authors did a great effort to make the methods section detailed and reproducible. The summary file (e.g. : 05_multi-IAA12_COM2.docx) is however not provided for all HADDOCK runs. This file is essential to assess how many structures are present in the cluster, since a low amount of structures in a defined cluster might drive to misinterpretation in the following steps.

In Figure 6, the legend (line 1180) states: "Time evolution of instantaneous ΔG_{eff} values over 20 ns identifying stable TIR1·AUX/IAAPB1 complexes from HADDOCK best scoring groups.". And related to the same figure in the main text (line 375): "We used as a starting structure (t=0) the results from the HADDOCK simulations including the degron tail restraint.". As all models are now provided, the best scoring groups from HADDOCK should be mentioned in a way they can be retrieved in the supplementary data.

[3] DisVis-IAA07 (all-restraints) run was performed with only 4 restraints, one of which with a different minimum-maximum distance of 20 Å and 27.4 Å, respectively:

```
A 226 CA B 228 CA 8.0 34.0
A 229 CA B 228 CA 8.0 34.0
A 2013 CA B 124 CA 20.0 27.4
A 2013 CA B 242 CA 8.0 34.0
```

Same for the DisVis-IAA07 (all-restraints) run, an apparently arbitrary range of minimum-maximum distance seem to have been set (24.8 Å and 36.0 Å, respectively):

```
A 226 CA B 233 CA 8.0 34.0
A 67 CA B 233 CA 8.0 34.0
A 2013 CA B 121 CA 24.8 36.0
```

These distance ranges are narrower than the distance restraint set for the others, and from what is expected for the DSBU cross-linker itself (which should be 34 Å, as stated by the authors). The authors should clarify in the method section why a subset of restraints was used to run DisVis analysis, and how these have been selected since they are different from those provided in the Supplementary Tables 2 and 3.

Reviewer #5 (Remarks to the Author):

According to my opinion, the authors answered to the most of the remarks made by Reviewer#3. It is very good that the authors provided detailed description of the computational methods in the Supplement Info. However, I fully agree with the Reviewer#3 that a sentence or two regarding the

details of the MD simulations including thermostat, barostat, time step and ensemble should be added in the Methods part of the manuscript. Also, I am deeply concerned about the short simulation time of only 20 ns. At least RMSD graphs of the proteins' backbone during the MD simulations must be provided, either to convince the reader that the simulations were equilibrated, or to give a chance to the reader to judge by itself how equilibrated the systems are and how reliable are the computational results due to short simulation time. Altogether, it is an interested paper, but before publishing, two issues regarding the computational part should be done: 1) to add a few details regarding the computational procedure and 2) to add RMSD graphs of the simulations in manuscript.

Response to Reviewers:

Reviewer #4 (Remarks to the Author):

The authors have performed an impressive array of work for this rebuttal and I commend them on their efforts. The readability of the manuscript has improved significantly and most of the requests for clarification about the structural modelling addressed. In addition, the MS experiment are now replicated and the instrumentation used is now consistent across the different runs.

In my opinion there remain however three small points related to the modelling procedures that need to be addressed before publication, as I believe they are important to the work.

[1] In the file “haddock_files.tar\haddock_files\IAA07\No_disorder_restraint”, the PDB cluster 1_1 is missing. In the summary of the run it is indicated as present. It is an important point to clarify as this cluster is the most significant (124 of the 175 structures) and the first structure is usually the highest scoring one, so it should be included or its absence explained.

We agree with the reviewer, and apologize for the unintended omission. The missing files are now included in **haddock_files.tgz** file.

[2] The authors did a great effort to make the methods section detailed and reproducible. The summary file (e.g.: 05_multi-IAA12_COM2.docx) is however not provided for all HADDOCK runs. This file is essential to assess how many structures are present in the cluster, since a low amount of structures in a defined cluster might drive to misinterpretation in the following steps.

We thank the reviewer for the comment and found the summary files missing in the DisVis-assisted dockings only, which are not part of the main manuscript. The file (**disvis_haddock.tgz**) has been updated and contains the summary files in the respective subfolders named: “summary”.

In Figure 6, the legend (line 1180) states: “Time evolution of instantaneous ΔG_{eff} values over 20 ns identifying stable TIR1-AUX/IAAPB1 complexes from HADDOCK best scoring groups.”. And related to the same figure in the main text (line 375): “We used as a starting structure (t=0) the results from the HADDOCK simulations including the degon tail restraint.”. As all models are now provided, the best scoring groups from HADDOCK should be mentioned in a way they can be retrieved in the supplementary data.

We agree with the reviewer. We have now added information on the exact structures we used for each MD simulation in the Legend of Figure 6, and in the main text (now line: 373). In brief, we used the first structure from each cluster, namely: cluster1_1 and 2_1 for IAA7; and cluster1_1; 2_1; 3_1 for IAA12. We moved the detailed file description to the Methods section (line: 789). The minimized starting structures (t0) can also be found in the Supplementary file **MD.tgz**.

[3] DisVis-IAA07 (all-restraints) run was performed with only 4 restraints, one of which with a different minimum-maximum distance of 20 Å and 27.4 Å, respectively:

```
A 226 CA B 228 CA 8.0 34.0
A 229 CA B 228 CA 8.0 34.0
A 2013 CA B 124 CA 20.0 27.4
A 2013 CA B 242 CA 8.0 34.0
```

Same for the DisVis-IAA07 (all-restraints) run, an apparently arbitrary range of minimum-maximum distance seem to have been set (24.8 Å and 36.0 Å, respectively):

A 226 CA B 233 CA 8.0 34.0
A 67 CA B 233 CA 8.0 34.0
A 2013 CA B 121 CA 24.8 36.0

These distance ranges are narrower than the distance restraint set for the others, and from what is expected for the DSBU cross-linker itself (which should be 34 Å, as stated by the authors).

The authors should clarify in the method section why a subset of restraints was used to run DisVis analysis, and how these have been selected since they are different from those provided in the Supplementary Tables 2 and 3.

We again thank the reviewer for his/her attention to detail. We assume the reviewer meant DisVis-IAA12 in the second part of his/her statement. We have amended the Supplementary Table 2 reflecting these restraints. Of note, both restraints mentioned by the reviewer are not based on crosslinks, but on the theoretical stokes radii that can be occupied by the distinct disordered degron tails of either IAA7 or IAA12. Those distances were calculated according to Hamdi *et al.* as described in the Methods section. We have now included an additional sentence in the Methods section for clarity (line: 763).

Reviewer #5 (Remarks to the Author):

According to my opinion, the authors answered to the most of the remarks made by Reviewer#3. It is very good that the authors provided detailed description of the computational methods in the Supplement Info. However, I fully agree with the Reviewer#3 that a sentence or two regarding the details of the MD simulations including thermostat, barostat, time step and ensemble should be added in the Methods part of the manuscript.

We acknowledge the reviewer's comment, and have now moved the requested details from the Supplementary file to the Methods section in the Main manuscript (lines: 803-818).

Also, I am deeply concerned about the short simulation time of only 20 ns. At least RMSD graphs of the proteins' backbone during the MD simulations must be provided, either to convince the reader that the simulations were equilibrated, or to give a chance to the reader to judge by itself how equilibrated the systems are and how reliable are the computational results due to short simulation time.

We absolutely understand the concerns of the reviewer, and agree longer simulations would have been appropriate, if the systems would not have reached equilibrium. In our case however, data shows the simulations of TIR1-IAA7^{PB1} and TIR1-IAA12^{PB1} were equilibrated. We have followed the reviewer's advice and have added the RMSD graphs to Supplementary Figure 12 (panels c-d) for clarity. These show only minor movements for the backbones (<0.2 nm after approx. 6 ns) indicating those complexes are stable and simulations were equilibrated.

Altogether, it is an interested paper, but before publishing, two issues regarding the computational part should be done: 1) to add a few details regarding the computational procedure and 2) to add RMSD graphs of the simulations in manuscript.

We complied with the reviewer's suggestions. See all of the above.

REVIEWERS' COMMENTS:

Reviewer #4 (Remarks to the Author):

All comments were addressed in a satisfactory manner and I recommend publication.

Reviewer #5 (Remarks to the Author):

The authors successfully addressed both major points regarding the computational part raised in the previous round of review and I agree that the manuscript is accepted for publication.

Niemeyer et al.

Response to Reviewers:

Reviewer #4 (Remarks to the Author):

All comments were addressed in a satisfactory manner and I recommend publication.

We thank the reviewer for recommending our manuscript for publication. We value the reviewer's time and constructive feedback during the review process.

Reviewer #5 (Remarks to the Author):

The authors successfully addressed both major points regarding the computational part raised in the previous round of review and I agree that the manuscript is accepted for publication.

We thank the reviewer for recommending our manuscript for publication, and for providing valuable input in the last revision round.